# An assay for de novo kinetochore assembly reveals a key role for the CENP-T pathway in budding yeast

Jackie Lang[1,2], Adrienne Barber[1], Sue Biggins[1]*

[1]Division of Basic Sciences, Howard Hughes Medical Institute, Fred Hutchinson Cancer Research Center, Seattle, United States; [2]Molecular and Cellular Biology Program, University of Washington, Seattle, United States

**Abstract** Chromosome segregation depends on the kinetochore, the machine that establishes force-bearing attachments between DNA and spindle microtubules. Kinetochores are formed every cell cycle via a highly regulated process that requires coordinated assembly of multiple subcomplexes on specialized chromatin. To elucidate the underlying mechanisms, we developed an assay to assemble kinetochores de novo using centromeric DNA and budding yeast extracts. Assembly is enhanced by mitotic phosphorylation of the Dsn1 kinetochore protein and generates kinetochores capable of binding microtubules. We used this assay to investigate why kinetochores recruit the microtubule-binding Ndc80 complex via two receptors: the Mis12 complex and CENP-T. Although the CENP-T pathway is non-essential in yeast, we demonstrate that it becomes essential for viability and Ndc80c recruitment when the Mis12 pathway is crippled by defects in Dsn1 phosphorylation. Assembling kinetochores de novo in yeast extracts provides a powerful and genetically tractable method to elucidate critical regulatory events in the future.
DOI: https://doi.org/10.7554/eLife.37819.001

*For correspondence:
sbiggins@fredhutch.org

Competing interests: The authors declare that no competing interests exist.

## Introduction

Chromosomes must be accurately segregated to daughter cells during cell division to avoid aneuploidy, a hallmark of birth defects and cancers (*Pfau and Amon, 2012*). Faithful segregation relies on the attachment of chromosomes to spindle microtubules via the kinetochore, a conserved protein complex that assembles at centromeres (*Yamagishi et al., 2014*; *Musacchio and Desai, 2017*). Kinetochores must track dynamically growing and shrinking microtubule tips, monitor for erroneous kinetochore-microtubule attachments, and serve as the platform for the spindle assembly checkpoint (*Biggins, 2013*; *Joglekar and Kukreja, 2017*). To carry out these many functions, the kinetochore is a highly regulated, megadalton protein structure composed of many subcomplexes (*Figure 1A*). Although these subcomplexes must faithfully assemble onto the centromere every cell cycle, the underlying mechanisms that regulate kinetochore assembly are not well understood.

Kinetochores are built on a specialized centromeric chromatin structure, in which canonical histone H3 is replaced with a centromere-specific variant, CENP-A (*Earnshaw and Rothfield, 1985*; *Salmon and Bloom, 2017*). Most eukaryotes have complex 'regional' centromeres composed of repetitive DNA stretches with interspersed CENP-A- and H3-containing nucleosomes (*Blower et al., 2002*). The constitutive centromere-associated network (CCAN) binds to centromeric chromatin to form the inner kinetochore and serve as the scaffold for outer kinetochore assembly, which mediates microtubule attachment (*Foltz et al., 2006*). The Ndc80 complex ('Ndc80c') is a key microtubule-binding site within the kinetochore, because it directly mediates attachment and recruits additional attachment factors, such as the Ska complex in vertebrates and its functional ortholog Dam1 in fungi (*Cheeseman et al., 2006*; *DeLuca et al., 2005*; *Maure et al., 2011*; *Zhang et al., 2017*;

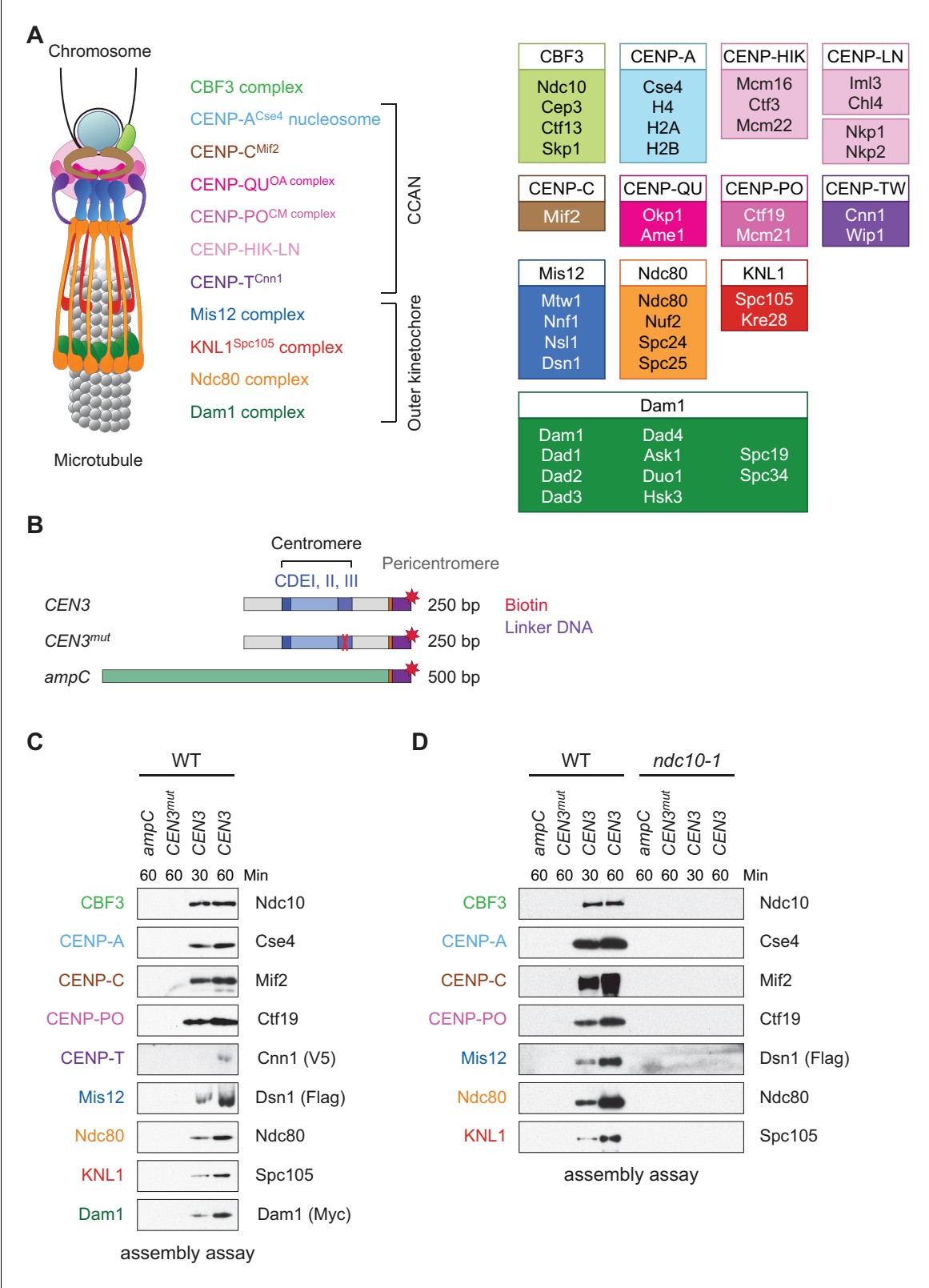

**Figure 1.** Assembly of kinetochores de novo on a centromere DNA template. (**A**) A schematic of the budding yeast kinetochore. Inner kinetochore subcomplexes assemble onto centromeres, serving as the platform for outer kinetochore recruitment. The listed subcomplexes are ordered based on physical interactions, and the yeast proteins in each kinetochore subcomplex are shown on the right. (**B**) DNA templates for the assembly assay. The templates include 500 bp from the *E. coli ampC* gene that encodes for β-lactamase (green) as a negative control, the 117 bp chromosome III

*Figure 1 continued on next page*

*Figure 1 continued*

centromere (*CEN3*), or a mutant *CEN3* (*CEN3^mut*) containing three point mutations in the CBF3 binding site (red 'X'). The three Centromere-Determining Elements (CDEs) are indicated and ~70 bp of flanking pericentromeric DNA on either side is shown (grey). The DNA templates also contain linker DNA (purple) before the biotinylation (red star) at the 3' end of the centromere. (C) Kinetochores assembled in vitro are centromere-specific and span the entire kinetochore. The indicated DNA templates were incubated in WT whole cell extracts prepared from a *CNN1-3V5 DSN1-3Flag DAM1-9myc* strain (SBY17228) for the indicated time (min). DNA-bound proteins were analyzed by immunoblotting with the indicated antibodies. Extracts are shown in *Figure 1—figure supplement 1*. (D) Kinetochore assembly is inhibited in an *ndc10-1* temperature sensitive mutant. Extracts from a *DSN1-6His-3Flag* strain (SBY8253) or a *DSN1-6His-3Flag ndc10-1* strain (SBY8361) shifted to the non-permissive temperature were used for assembly assays. DNA-bound proteins were analyzed by immunoblotting with the indicated antibodies. Extracts in *Figure 1—figure supplement 2*.
DOI: https://doi.org/10.7554/eLife.37819.002

The following figure supplements are available for figure 1:

**Figure supplement 1.** Whole cell extracts (left) and assembled kinetochores (right) from *Figure 1C*, immunoblotted with the indicated antibodies. * indicates a background band in the extract in all figures.
DOI: https://doi.org/10.7554/eLife.37819.003

**Figure supplement 2.** Whole cell extracts (left) and assembled kinetochores (right) from *Figure 1D*, immunoblotted with the indicated antibodies.
DOI: https://doi.org/10.7554/eLife.37819.004

*Lampert et al., 2013*). Interestingly, there are two parallel kinetochore receptors for Ndc80c: the Mis12 complex ('Mis12c') and CENP-T (*Maskell et al., 2010*; *Malvezzi et al., 2013*; *Schleiffer et al., 2012*; *Gascoigne et al., 2011*; *Nishino et al., 2013*; *Nishino et al., 2012*). Mis12c interacts with the KNL1 and Ndc80 complexes to create a larger network called KMN (KNL1-Mis12-Ndc80) (*Cheeseman et al., 2006*). CENP-T, a histone-fold domain containing protein, recruits Ndc80c via the same interaction surface on the Ndc80 complex that binds Mis12c (*Nishino et al., 2013*; *Schleiffer et al., 2012*; *Malvezzi et al., 2013*; *Hori et al., 2008*; *Dimitrova et al., 2016*). Although Mis12c and CENP-T each contribute to Ndc80c recruitment in vivo (*Malvezzi et al., 2013*; *Gascoigne et al., 2011*), it has been unclear why cells employ two competing receptors for Ndc80c and whether the CENP-T protein functions as a histone at centromeres.

Substantial progress in understanding kinetochore assembly has been made using reconstitution systems in vitro. For example, pre-formed nucleosomal arrays incubated in *Xenopus* egg extracts assemble microtubule-binding elements that allowed the identification of events required to initiate kinetochore assembly (*Guse et al., 2011*). Furthermore, the binding selectivity of some kinetochore proteins for CENP-A nucleosomes (over H3 nucleosomes) was recently determined by reconstituting the entire linkage between the CENP-A nucleosome and KMN (*Weir et al., 2016*). To identify additional events that regulate kinetochore assembly, we set out to develop a reconstitution system that combines the strengths of these previously developed methods with the added ability to genetically manipulate the system and maintain post-translational modifications. To do this, we used budding yeast because they have a simple 'point' centromere that is defined by a ~ 125 bp specific DNA sequence and a single microtubule attachment site per chromosome (*Winey et al., 1995*; *Biggins, 2013*). The kinetochore subcomplexes and functions are largely conserved, including the specialized chromatin structure containing CENP-A^Cse4 that serves as the platform for kinetochore assembly. Additionally, dual pathways for Ndc80 recruitment are used in yeast, although the CENP-T ortholog, Cnn1, is not essential for viability (*Schleiffer et al., 2012*; *Bock et al., 2012*). However, CENP-T^Cnn1 contributes to kinetochore function because mutants display increased chromosome loss, and the tethering of CENP-T^Cnn1 imparts partial stability to acentric minichromosomes via Ndc80 recruitment (*Malvezzi et al., 2013*; *Bock et al., 2012*). In yeast, CENP-T^Cnn1 localization to kinetochores peaks in mitosis as a result of phosphoregulation (*Schleiffer et al., 2012*; *Bock et al., 2012*), but it is unclear why its recruitment is cell cycle regulated and whether this affects the Mis12 pathway of Ndc80 recruitment.

Here, we develop a cell-free method to assemble complete kinetochores de novo using centromeric DNA and yeast extracts. We demonstrate that this assay has the same basic requirements as kinetochore assembly in vivo, including the need for the CENP-A^Cse4 chaperone HJURP^Scm3, suggesting the formation of a centromeric nucleosome. Conserved mitotic phosphorylation events of the Mis12 complex enhance kinetochore assembly, revealing that the assay is sensitive to key post-translational modifications. Furthermore, this method generates kinetochores that exhibit microtubule-binding activity and employ both Ndc80 recruitment pathways. We applied this assay to

identify the requirements for CENP-T[Cnn1] assembly and find that it requires all other inner kineto-chore subcomplexes, suggesting it does not have independent DNA-binding activity. Furthermore, we discovered that the CENP-T[Cnn1] pathway is required for Ndc80 recruitment and cell viability when the Mis12 pathway is impaired by defects in conserved mitotic phosphoregulation (*Akiyoshi et al., 2013a*; *Kim and Yu, 2015*; *Yang et al., 2008*). Taken together, we have established a kinetochore assembly assay that identifies a critical function for the yeast CENP-T[Cnn1] pathway and that provides a powerful method to identify other key regulatory events required for kinetochore assembly and function in the future.

## Results

### Development of a method to assemble kinetochores de novo

Because we had previously identified conditions to purify functional kinetochores from yeast cells (*Akiyoshi et al., 2010*), we reasoned that these extracts might be permissive for de novo kineto-chore assembly. To test this, we linked the chromosome III centromere (117 bp) and ~70 bp of peri-centromeric DNA on each side (referred to as 'CEN3'; *Figure 1B*) to beads via a biotin tag and incubated it in whole cell yeast extracts in the presence of excess non-specific competitive DNA. As negative controls, we used a template (CEN3[mut]) with mutations in the centromere determining ele-ment III (CDEIII) region of DNA that abolishes kinetochore assembly in vivo (*Sorger et al., 1995*; *Lechner and Carbon, 1991*) as well as a 500 bp DNA template from within the *E. coli ampC* gene. We optimized the extract conditions for assembly in vitro by altering the lysis buffer and method, most notably switching from potassium chloride to potassium glutamate, a salt utilized in other reconstitution assays (*Seki and Diffley, 2000*; *Heller et al., 2011*). The assembly reaction was ini-tially performed using an extract prepared from asynchronously growing wildtype (WT) cells and analyzed by immunoblotting against representative components of most kinetochore subcomplexes. Within 30 min of assembly, every protein assayed bound specifically to centromeric DNA (*Figure 1C*). Inner kinetochore components are generally saturated within 30 min, while outer kineto-chore proteins require longer to reach saturation. To compare the efficiency of assembly in various mutants and conditions, we analyzed assembly on CEN3 DNA at two time points hereafter.

To further analyze the composition of the assembled particles, we performed mass spectrometry. We detected 39 out of 49 core kinetochore proteins at higher coverage levels on CEN3 DNA rela-tive to either *ampC* DNA or CEN3[mut] DNA (*Table 1*). In support of centromeric nucleosome assem-bly, CENP-A[Cse4] was specifically enriched on CEN3 DNA. Importantly, we detected components from all known kinetochore subcomplexes on CEN3 DNA, including the CENP-T[Cnn1] protein. The only proteins that were not detected are small proteins that are components of subcomplexes that were otherwise detected in the MS (for example, Dad2 in the Dam1 complex). Together, these data suggest that all kinetochore complexes assemble on centromeric DNA under the conditions we developed.

We next asked whether the assembly assay reflected requirements in vivo. Kinetochore assembly is initiated by the binding of the CBF3 complex to CDEIII, which facilitates the deposition of CENP-A[Cse4] (*Poddar et al., 2004*; *Camahort et al., 2007*). All kinetochore proteins except Cbf1, which binds directly to CDEI, require the Ndc10 component of the CBF3 complex for their localization in vivo (*He et al., 2000*). We therefore tested the requirement for CBF3 by performing the assembly assay with extracts prepared from WT cells and an *ndc10-1* temperature sensitive mutant. Similar to the negative controls, the assembly reaction was completely inhibited on the CEN3 DNA in the *ndc10-1* extracts (*Figure 1D*). Together, these data indicate that the assembly reaction is initiated by CBF3, consistent with the requirements for assembly in vivo.

### Kinetochores assemble on a single CENP-A nucleosome

Kinetochore assembly in vivo requires a CENP-A nucleosome, so we tested whether CENP-A[Cse4] requires its chaperone HJURP[Scm3] for deposition (*Camahort et al., 2007*; *Shivaraju et al., 2011*; *Stoler et al., 2007*). To do this, we generated cells containing an auxin-inducible degron (AID) allele of *SCM3*, *scm3-AID*, which targets the protein for proteasomal degradation when the TIR1 F-box protein and the hormone auxin are present (*Nishimura et al., 2009*). Although we could not detect HJURP[Scm3] protein in extracts due to low intracellular levels, we concluded that the protein was

**Table 1.** Components from each of the core subcomplexes are detected on assembled kinetochores.

Kinetochores were assembled on *ampC*, *CEN3^mut*, or *CEN3* DNA from an asynchronous WT *DSN1-3Flag* (SBY14441) extract and analyzed by LC/MS/MS mass spectrometry. The table indicates the human ortholog (if applicable) of each yeast protein, the percent coverage, and the number of unique and total peptides detected from each assembly. We included the only detected microtubule-associated protein.

**Table 1. WT assembled kinetochores**

| | Subcomplex | Yeast Protein | Human Protein | *ampC* % Coverage | *ampC* Unique Peptides | *ampC* Total Peptides | *CEN3^mut* % Coverage | *CEN3^mut* Unique Peptides | *CEN3^mut* Total Peptides | *CEN3* % Coverage | *CEN3* Unique Peptides | *CEN3* Total Peptides |
|---|---|---|---|---|---|---|---|---|---|---|---|---|
| | CPC | Ipl1 | Aurora B | Not present | | | Not present | | | 23.7 | 8 | 10 |
| | | Sli15 | INCENP | 7 | 2 | 2 | 14.8 | 6 | 7 | 64.3 | 54 | 113 |
| | | Bir1 | Survivin | 15.1 | 10 | 11 | 22.6 | 14 | 17 | 59.2 | 70 | 177 |
| | | Nbl1 | Borealin | Not present | | | Not present | | | 76.7 | 6 | 11 |
| CCAN | Cbf1 | Cbf1 | Cbf1 | Not present | | | 62.7 | 30 | 77 | 59.3 | 29 | 60 |
| | Cbf3 | Ndc10 | | 11.4 | 7 | 8 | 32.9 | 23 | 26 | 63.2 | 78 | 194 |
| | | Cep3 | | 3.5 | 1 | 2 | 15.3 | 8 | 9 | 34.2 | 25 | 76 |
| | | Ctf13 | | 2.7 | 1 | 1 | 18.4 | 5 | 5 | 46 | 22 | 38 |
| | | Skp1 | | 28.4 | 3 | 3 | 19.6 | 2 | 2 | 41.8 | 12 | 24 |
| | Nucleosome | Cse4 | CENP-A | 13.1 | 3 | 3 | 24.5 | 6 | 8 | 49.8 | 10 | 31 |
| | | Hta2 | H2A | 35.6 | 7 | 20 | 35.6 | 5 | 13 | 35.6 | 6 | 19 |
| | | Htb2 | H2B | 45 | 8 | 18 | 39.7 | 7 | 29 | 39.7 | 7 | 29 |
| | | Hht1 | H3 | 5.1 | 1 | 1 | 5.1 | 1 | 1 | Not present | | |
| | | Hhf1 | H4 | 45.6 | 7 | 11 | 56.3 | 8 | 19 | 46.6 | 8 | 13 |
| | Nucleosome | Psh1 | | 3.9 | 1 | 1 | Not present | | | Not present | | |
| | Associated | Scm3 | HJURP | Not present | | | 6.3 | 1 | 1 | 28.3 | 8 | 10 |
| | Mif2 | Mif2 | CENP-C | Not present | | | 9.7 | 4 | 4 | 58.7 | 29 | 39 |
| | OA | Okp1 | CENP-Q | Not present | | | 20.9 | 7 | 9 | 42.6 | 21 | 34 |
| | | Ame1 | CENP-U | Not present | | | 20.4 | 5 | 6 | 61.4 | 22 | 41 |
| | CM | Ctf19 | CENP-P | Not present | | | 6.8 | 2 | 2 | 44.7 | 21 | 31 |
| | | Mcm21 | CENP-O | Not present | | | 10.3 | 3 | 4 | 65.8 | 26 | 39 |
| | Iml3 | Iml3 | CENP-L | Not present | | | Not present | | | 60.8 | 13 | 19 |
| | | Chl4 | CENP-N | Not present | | | 7.9 | 3 | 3 | 37.1 | 16 | 19 |
| | | Nkp1 | | Not present | | | 26.9 | 4 | 6 | 57.6 | 17 | 28 |
| | | Nkp2 | | Not present | | | 15 | 2 | 4 | 55.6 | 7 | 10 |
| | Ctf3 | Mcm16 | CENP-H | Not present | | | 22.7 | 2 | 3 | 48.6 | 7 | 11 |
| | | Ctf3 | CENP-I | Not present | | | 5.3 | 3 | 3 | 23.7 | 17 | 27 |

*Table 1 continued on next page*

Table 1 continued

**Table 1.** WT assembled kinetochores

| | | | | ampC | ampC | ampC | CEN3$^{mut}$ | CEN3$^{mut}$ | CEN3$^{mut}$ | CEN3 | CEN3 | CEN3 |
|---|---|---|---|---|---|---|---|---|---|---|---|---|
| | | Mcm22 | CENP-K | Not present | | | 25.5 | 3 | 4 | 81.6 | 18 | 27 |
| | Cnn1 | Cnn1 | CENP-T | Not present | | | Not present | | | 45.7 | 13 | 18 |
| | | Wip1 | CENP-W | Not present | | | Not present | | | 39.3 | 2 | 2 |
| | | Mhf1 | CENP-S | 48.9 | 4 | 4 | 48.9 | 3 | 7 | 40 | 2 | 5 |
| | | Mhf2 | CENP-X | 43.8 | 4 | 8 | 47.5 | 4 | 6 | 28.8 | 3 | 4 |
| Outer Kt | Mtw1 | Mtw1 | Mis12 | Not present | | | Not present | | | 22.8 | 4 | 4 |
| | | Nnf1 | PMF1 | Not present | | | Not present | | | 13.9 | 2 | 2 |
| | | Nsl1 | Nsl1 | Not present | | | Not present | | | 24.1 | 3 | 3 |
| | | Dsn1 | Dsn1 | Not present | | | Not present | | | 7.3 | 2 | 2 |
| | Ndc80 | Ndc80 | HEC1 | Not present | | | Not present | | | 28.4 | 15 | 16 |
| | | Nuf2 | NUF2 | Not present | | | Not present | | | 32.8 | 12 | 13 |
| | | Spc24 | SPC24 | Not present | | | Not present | | | 57.3 | 8 | 8 |
| | | Spc25 | SPC25 | Not present | | | Not present | | | 27.1 | 5 | 5 |
| | Spc105 | Spc105 | KNL1 | Not present | | | Not present | | | 5 | 3 | 3 |
| | | Kre28 | Zwint1 | Not present | | | Not present | | | Not present | | |
| | Dam1 | Dam1 | | Not present | | | Not present | | | 10.8 | 2 | 2 |
| | | Dad1 | | 26.6 | 1 | 1 | 26.6 | 1 | 2 | 26.6 | 1 | 1 |
| | | Dad3 | | Not present | | | Not present | | | Not present | | |
| | | Ask1 | | Not present | | | Not present | | | 8.2 | 1 | 1 |
| | | Duo1 | | Not present | | | Not present | | | 7.3 | 1 | 1 |
| | | Hsk3 | | 15.9 | 1 | 1 | 15.9 | 1 | 1 | 15.9 | 1 | 1 |
| | | Spc19 | | Not present | | | Not present | | | 8.5 | 1 | 1 |
| | | Spc34 | | Not present | | | Not present | | | 4.7 | 1 | 2 |
| | | Dad2 | | Not present | | | Not present | | | Not present | | |
| | | Dad4 | | Not present | | | Not present | | | Not present | | |
| | MAPs | Stu2 | CHTOG | 3.5 | 2 | 2 | Not present | | | Not present | | |

DOI: https://doi.org/10.7554/eLife.37819.005

degraded because the cells were inviable when plated on auxin (*Figure 2—figure supplement 1*). We prepared extracts from *scm3-AID* strains (with or without *TIR1*) treated with auxin and performed the assembly assay (*Figure 2A*). As expected for the most upstream protein in the assembly pathway, Ndc10 associated with *CEN3* DNA in the presence or absence of HJURP$^{Scm3}$. However, CENP-A$^{Cse4}$ and all other CCAN components assayed no longer associated with *CEN3* when HJURP$^{Scm3}$ was depleted (*Figure 2A*). This strict requirement for CENP-A$^{Cse4}$ recruitment by its chaperone suggests that CENP-A$^{Cse4}$ is forming a functional nucleosome in vitro.

Centromeric nucleosomes can be detected in the surrounding pericentromeric region in vivo (*Coffman et al., 2011*; *Lawrimore et al., 2011*; *Wisniewski et al., 2014*), leading to debate about whether a single CENP-A$^{Cse4}$ nucleosome is sufficient for kinetochore assembly (*Aravamudhan et al., 2013*; *Furuyama and Biggins, 2007*; *Wisniewski et al., 2014*). We therefore performed the assembly assay using a shorter 180 bp template that cannot form more than one octameric nucleosome. CENP-A$^{Cse4}$ levels were similar on both templates, and the entire kinetochore formed in both cases (*Figure 2B*). Together, these data suggest that a single, well-positioned

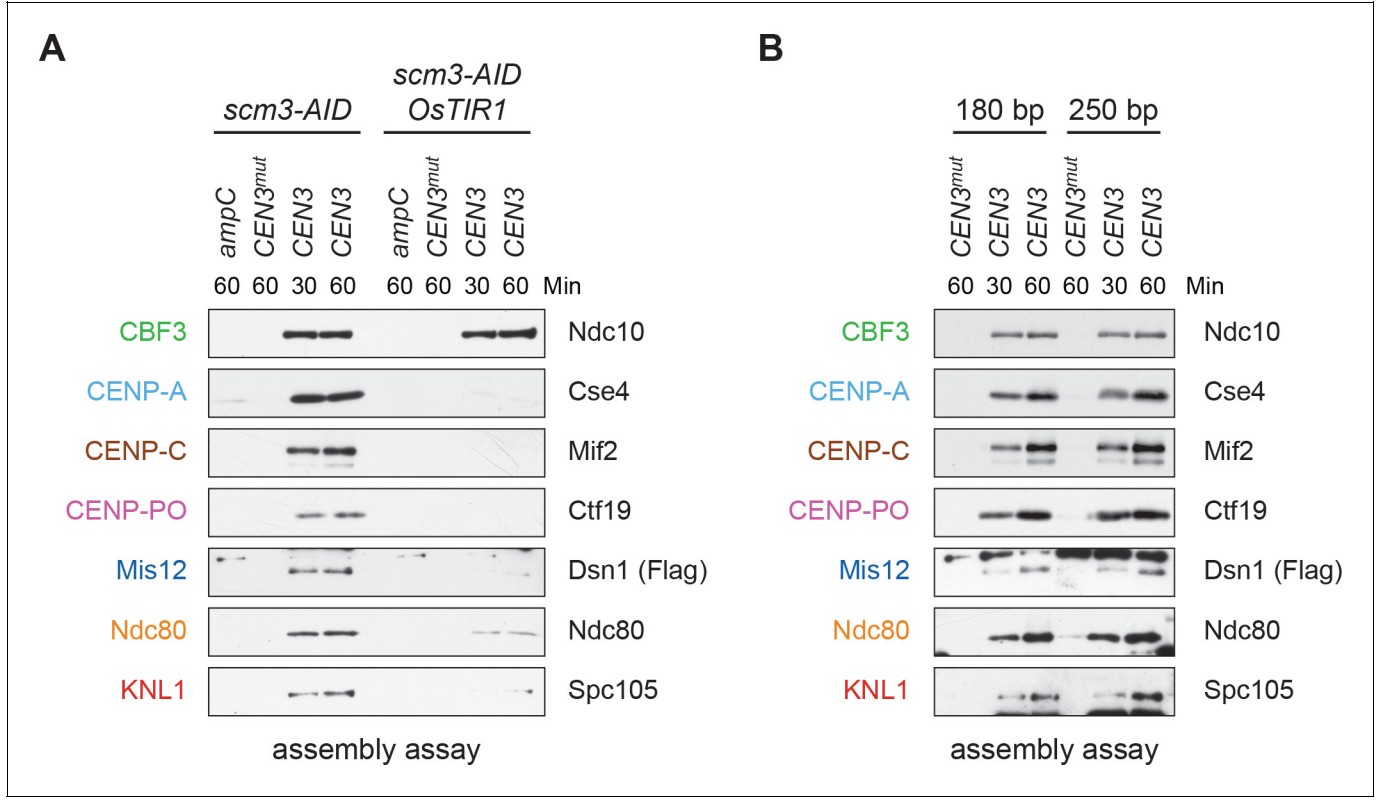

**Figure 2.** Assembled kinetochore particles contain a single, chaperone-dependent CENP-A$^{Cse4}$ nucleosome. (**A**) Degradation of HJURP$^{Scm3}$ blocks assembly of the kinetochore beginning with CENP-A$^{Cse4}$. A *DSN1-3Flag scm3-EGFP-AID* strain (SBY16440) and a *DSN1-3Flag scm3-EGFP-AID OsTIR1-myc* strain (SBY16438) were treated with auxin and the extracts were used for assembly assays. DNA-bound proteins were analyzed by immunoblotting for the indicated proteins. Extracts in *Figure 2—figure supplement 2*. (**B**) Assembly on a centromeric DNA template of only 180 bp is similar to a 250 bp template. Extract from a *DSN1-3Flag CNN1-3V5 DAM1-9myc* (SBY17228) strain was used for assembly assays with the indicated DNA templates. DNA-bound proteins were analyzed by immunoblotting with the indicated antibodies. Extracts in *Figure 2—figure supplement 3*.
DOI: https://doi.org/10.7554/eLife.37819.006

The following figure supplements are available for figure 2:

**Figure supplement 1.** Degradation of HJURP$^{Scm3}$ is lethal to cells.
DOI: https://doi.org/10.7554/eLife.37819.007

**Figure supplement 2.** Whole cell extracts (left) and assembled kinetochores (right) from *Figure 2A*, immunoblotted with the indicated antibodies.
DOI: https://doi.org/10.7554/eLife.37819.008

**Figure supplement 3.** Whole cell extracts (left) and assembled kinetochores (right) from *Figure 2B*, immunoblotted with the indicated antibodies.
DOI: https://doi.org/10.7554/eLife.37819.009

centromeric nucleosome is sufficient for kinetochore assembly in the absence of surrounding peri-centromeric DNA.

## Assembly in vitro is regulated by the cell cycle and phosphorylation

Kinetochore assembly is regulated during the cell cycle and occurs during S phase in budding yeast (*Kitamura et al., 2007*; *Pearson et al., 2004*), although it isn't clear whether this reflects a requirement for active DNA replication or another S phase event. During mitosis, there are dynamic changes in kinetochore composition and CENP-T$^{Cnn1}$ levels at kinetochores peak due to phosphoregulation (*Schleiffer et al., 2012*; *Bock et al., 2012*; *Dhatchinamoorthy et al., 2017*). To test whether the assembly assay is subject to cell cycle regulation, WT cells were grown asynchronously or arrested in G1, S phase, or mitosis, and the extracts were used for in vitro assembly assays. Assembly is least efficient in extracts from cells arrested in G1 and most efficient in S phase and mitosis (*Figure 3A*), consistent with cell cycle regulation that occurs in vivo. As expected, the CENP-T$^{Cnn1}$ pathway is noticeably enhanced in kinetochores assembled from mitotic extracts. The cellular levels of some proteins, particularly CENP-T$^{Cnn1}$ and CENP-C$^{Mif2}$, are different in the various arrests and it is not clear whether this is due to changes in expression level or solubility. We also note that there appears to be a preference for the assembly of slower-migrating forms of CENP-C$^{Mif2}$ during S phase and mitosis, which may reflect post-translational modifications.

A conserved mitotic phosphorylation event that promotes kinetochore assembly is Aurora B-mediated phosphorylation of Dsn1, which promotes the interaction between Mis12c and the inner kinetochore protein CENP-C$^{Mif2}$ (*Akiyoshi et al., 2013a*; *Kim and Yu, 2015*; *Yang et al., 2008*; *Dimitrova et al., 2016*; *Petrovic et al., 2016*). To test the effects of this phosphorylation on kinetochore assembly in vitro, we made extracts made from a *dsn1-S240D, S250D* (*dsn1-2D*) phosphomimetic mutant (*Akiyoshi et al., 2013a*). While the innermost CCAN proteins were present at equivalent levels, the *dsn1*-2D assembled kinetochores showed a strong enrichment for outer kinetochore proteins beginning with the Mis12 complex itself when assayed by immunoblotting and mass spectrometry (*Figure 3B* and *Table 2*). To directly quantify the difference between WT and *dsn1-2D* assembly reactions, we performed quantitative mass spectrometry (qMS) using tandem mass tag labeling (*McAlister et al., 2014*). Although the qMS data does not allow us to analyze the stoichiometry of components within one sample, we were able to compare relative protein levels between WT and *dsn1-2D* assembled kinetochores. Similar to the immunoblot analysis, there was a strong enrichment of outer kinetochore proteins (3- to 7-fold enrichment) in the *dsn1-2D* assembled kinetochores while the CCAN levels were similar to WT assembled kinetochores (*Figure 3C*). Together, these data indicate that our assembly assay in vitro reflects requirements known for assembly in vivo and is sensitive to critical post-translational modifications.

## Assembled kinetochores are capable of binding microtubules

One of the most fundamental activities of the kinetochore is microtubule binding, so we next tested whether the assembled kinetochores are competent to attach to microtubules. We assembled kinetochores in extracts made from *dsn1-2D* cells, as well as extracts depleted of the major microtubule-binding component Ndc80 as a control (*Cheeseman et al., 2006*; *DeLuca et al., 2005*). Because the Chromosomal Passenger Complex (CPC) can mediate microtubule binding when bound to centromeric DNA in vitro (*Sandall et al., 2006*), we also performed the experiment in extracts depleted of INCENP$^{Sli15}$, the CPC scaffold protein (*Jeyaprakash et al., 2007*; *Carmena et al., 2012*). Although the AID-tagged proteins were significantly depleted from cells after auxin addition (*Figure 4A*), low levels of Ndc80 remained that were capable of assembling in vitro (*Figure 4B*). However, the residual levels were not sufficient for the proteins to perform their recruitment functions since Dam1 was absent in the *ndc80-AID* assembled kinetochores and Aurora B$^{Ipl1}$ was absent from the *sli15-AID* assembled kinetochores (*Jeyaprakash et al., 2007*; *Klein et al., 2006*; *Lampert et al., 2013*). We incubated the bead-bound assembled kinetochores with either taxol-stabilized microtubules or free tubulin as a negative control. The bound proteins were eluted from the beads and copurifying tubulin was analyzed by fluorescence. Microtubules bound much more robustly than free tubulin to the assembled *dsn1-2D* kinetochores (*Figure 4B*). Although the single mutants did not significantly alter microtubule binding, the double *ndc80-AID sli15-AID* mutant kinetochores were not able to bind

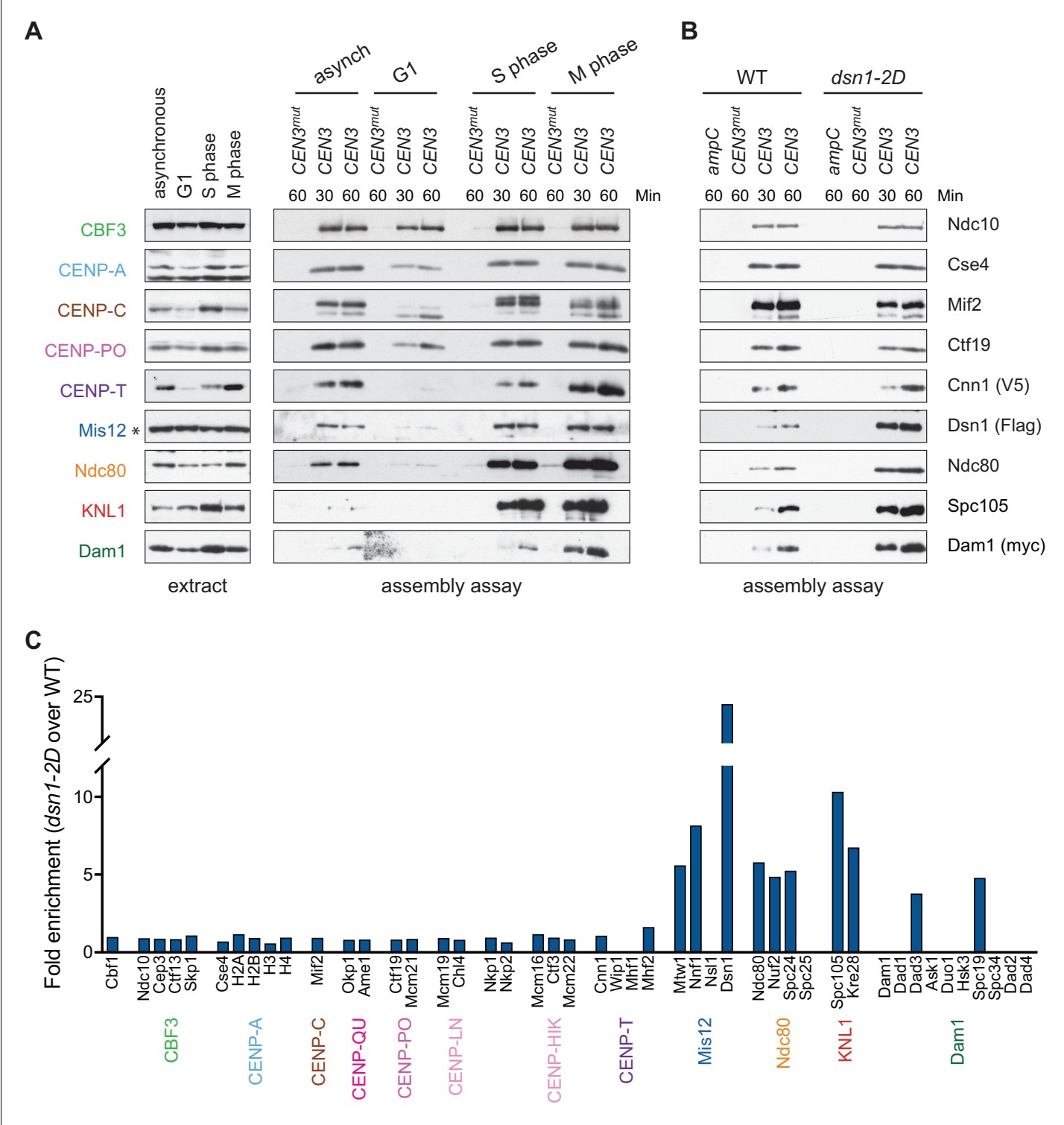

**Figure 3.** Kinetochore assembly in vitro is regulated by the cell cycle and phosphorylation. (A) Assembly in vitro is most efficient in extracts made from mitotically arrested cells. Kinetochores were assembled using extract from WT cells (*DSN1-3Flag CNN1-3V5 DAM1-9myc* (SBY17227)) that were either asynchronously growing or arrested in G1 (with alpha factor), S phase (with hydroxyurea), or early mitosis (with benomyl). Diluted whole cell extracts (left) and DNA-bound proteins (right) were analyzed by immunoblotting with the indicated antibodies. (B) Outer kinetochore assembly is enhanced in *dsn1-2D* extracts. Assembly assays were performed using extracts prepared from benomyl-arrested *DSN1-3Flag CNN1-3V5 DAM1-9myc* (SBY17228) and *dsn1-2D-3Flag CNN1-3V5 DAM1-9myc* (SBY17234) strains on the indicated DNA templates. DNA-bound proteins were analyzed by immunoblotting with the indicated antibodies. Extracts in *Figure 3—figure supplement 1*. (C) *dsn1-2D* enhances the assembly of most outer

*Figure 3 continued on next page*

*Figure 3 continued*

kinetochore proteins by at least 5-fold. Assembly assays were performed using extracts from *DSN1-3Flag* (SBY14441) and *dsn1-2D-3Flag* (SBY14151) on *CEN3* DNA. Assembled proteins were labeled with tandem mass tags and analyzed by quantitative mass spectrometry. For each protein, the relative abundance in *dsn1-2D* assembled kinetochores was divided by the relative abundance in WT to calculate the fold enrichment in the *dsn1-2D* assembled kinetochores.

DOI: https://doi.org/10.7554/eLife.37819.010

The following figure supplement is available for figure 3:

**Figure supplement 1.** Whole cell extracts (left) and assembled kinetochores (right) from *Figure 3B*, immunoblotted with the indicated antibodies.
DOI: https://doi.org/10.7554/eLife.37819.011

microtubules. Thus, kinetochores assembled in vitro are capable of binding to microtubules through the established microtubule-binding interfaces.

## CENP-T$^{Cnn1}$ requires all CCAN proteins for kinetochore localization

A conserved feature of kinetochore assembly is the recruitment of Ndc80 via two complexes: Mis12c and CENP-T$^{Cnn1}$. The mechanism of CENP-T$^{Cnn1}$ recruitment to the kinetochore has been controversial. CENP-T$^{Cnn1}$ and its partner CENP-W have histone fold domains (HFD) and can tetramerize with two additional HFD proteins, CENP-S/X, to form a nucleosome-like structure in vitro (*Nishino et al., 2012*; *Takeuchi et al., 2014*; *Schleiffer et al., 2012*). The human proteins require their heterotetramerization and DNA-binding capabilities to assemble a functional kinetochore in vivo (*Nishino et al., 2012*), leading to the model that CENP-T$^{Cnn1}$ forms a unique centromeric chromatin structure. However, CENP-T$^{Cnn1}$ localization to kinetochores requires other CCAN proteins, suggesting it may not directly bind to centromeric DNA (*Carroll et al., 2010*; *Basilico et al., 2014*; *Samejima et al., 2015*; *Suzuki et al., 2015*; *Pekgöz Altunkaya et al., 2016*; *Logsdon et al., 2015*).

To address these issues, we used our DNA-based method to analyze the requirements for CENP-T$^{Cnn1}$ recruitment. The CCAN is composed of distinct subcomplexes and the physical interactions between them have been mapped using co-immunoprecipitation experiments (*Schleiffer et al., 2012*; *Pekgöz Altunkaya et al., 2016*) (*Figure 1A*). To map the CENP-T$^{Cnn1}$ assembly pathway on centromeric DNA, we performed the assembly assay from cells containing a representative mutant of each conserved CCAN subcomplex that had been arrested in mitosis. CENP-T$^{Cnn1}$ is absent or severely reduced in all inner kinetochore mutants tested (*Figure 5A–C*), indicating that CENP-T$^{Cnn1}$ does not have independent DNA-binding properties in our assay conditions. In addition, CENP-T$^{Cnn1}$ appears to be the most distal component of the CCAN because every other subcomplex is required for its kinetochore localization (*Figure 5D*) (*Pekgöz Altunkaya et al., 2016*).

## Kinetochore assembly in vitro utilizes both pathways to Ndc80 recruitment

It has been unclear how CENP-T$^{Cnn1}$ facilitates kinetochore assembly in yeast, because Ndc80 levels are not noticeably reduced by the loss of CENP-T$^{Cnn1}$in vivo (*Bock et al., 2012*; *Schleiffer et al., 2012*). We therefore performed the assembly assay in a *cnn1Δ* extract and found that Ndc80 and KNL1$^{Spc105}$ levels are slightly reduced, suggesting that CENP-T$^{Cnn1}$ contributes to Ndc80 recruitment (*Figure 6*). We next compared this to the role of Mis12c in Ndc80 recruitment by performing the assay in an extract from which *dsn1-AID* had been degraded. Here, Ndc80 assembly is considerably reduced but not absent. To test whether the residual Ndc80 is due to CENP-T$^{Cnn1}$, we analyzed assembly from a *cnn1Δ dsn1-AID* double mutant and found that Ndc80 recruitment is abolished. Together, these data show that the de novo assay uses both pathways of assembly and that CENP-T$^{Cnn1}$ contributes to Ndc80 recruitment independently of the Mis12 complex.

## CENP-T$^{Cnn1}$ is essential for Ndc80 recruitment when Mis12c lacks mitotic phosphorylation

Although CENP-T$^{Cnn1}$ contributes to Ndc80 recruitment, Mis12c is clearly the major Ndc80 receptor in budding yeast (*Schleiffer et al., 2012*; *Bock et al., 2012*). Consistent with this, Mis12c is essential for viability while CENP-T$^{Cnn1}$ is non-essential, leading to the question of why yeast have retained the CENP-T$^{Cnn1}$ pathway. We hypothesized that CENP-T$^{Cnn1}$ may compensate for downregulation of

**Table 2.** Outer kinetochore assembly is enhanced by Dsn1 phosphorylation.

Kinetochores were assembled on the indicated DNA templates from an asynchronous *dsn1-2D-3Flag* (SBY14151) extract and analyzed by mass spectrometry as in *Table 1*. We included the detected microtubule-associated proteins.

**Table 2. dsn1-2D assembled kinetochores**

| | Subcomplex | Yeast Protein | Human Protein | *ampC* % Coverage | *ampC* Unique Peptides | *ampC* Total Peptides | CEN3^mut % Coverage | CEN3^mut Unique Peptides | CEN3^mut Total Peptides | CEN3 % Coverage | CEN3 Unique Peptides | CEN3 Total Peptides |
|---|---|---|---|---|---|---|---|---|---|---|---|---|
| | CPC | Ipl1 | Aurora B | Not present | | | Not present | | | 24.5 | 9 | 15 |
| | | Sli15 | INCENP | 11 | 4 | 5 | 21.9 | 10 | 11 | 62.6 | 61 | 221 |
| | | Bir1 | Survivin | 20.2 | 13 | 15 | 25.5 | 17 | 20 | 64.3 | 71 | 257 |
| | | Nbl1 | Borealin | 23.3 | 1 | 1 | 21.9 | 1 | 1 | 61.6 | 7 | 19 |
| CCAN | Cbf1 | Cbf1 | | Not present | | | 65 | 27 | 53 | 59.3 | 30 | 92 |
| | Cbf3 | Ndc10 | | 21.2 | 2 | 14 | 24.5 | 19 | 22 | 58.9 | 68 | 563 |
| | | Cep3 | | 20.6 | 8 | 9 | 15.3 | 8 | 10 | 34.5 | 21 | 111 |
| | | Ctf13 | | 2.7 | 1 | 1 | 12.3 | 5 | 5 | 40.6 | 21 | 71 |
| | | Skp1 | | 8.8 | 1 | 2 | 22.2 | 3 | 3 | 44.3 | 11 | 32 |
| | Nucleosome | Cse4 | CENP-A | 20.5 | 5 | 5 | 14 | 4 | 6 | 49.3 | 10 | 34 |
| | | Hta2 | H2A | 35.6 | 5 | 13 | 35.6 | 7 | 20 | 35.6 | 5 | 34 |
| | | Htb2 | H2B | 39.7 | 7 | 18 | 39.7 | 7 | 31 | 31.3 | 6 | 29 |
| | | Hht1 | H3 | Not present | | | 5.1 | 1 | 1 | Not present | | |
| | | Hhf1 | H4 | 56.3 | 9 | 11 | 56.3 | 8 | 16 | 55.3 | 8 | 22 |
| | Nucleosome Associated | Psh1 | | 7.4 | 2 | 2 | Not present | | | Not present | | |
| | | Scm3 | HJURP | Not present | | | Not present | | | 30.5 | 8 | 17 |
| | Mif2 | Mif2 | CENP-C | Not present | | | 13.7 | 5 | 5 | 55.7 | 27 | 54 |
| | OA | Okp1 | CENP-Q | Not present | | | 22.7 | 8 | 9 | 43.6 | 21 | 50 |
| | | Ame1 | CENP-U | Not present | | | 28.7 | 5 | 5 | 54.9 | 19 | 43 |
| | CM | Ctf19 | CENP-P | Not present | | | 9.8 | 3 | 3 | 42.3 | 17 | 41 |
| | | Mcm21 | CENP-O | Not present | | | 25.8 | 7 | 8 | 48.6 | 23 | 42 |
| | Iml3 | Iml3 | CENP-L | Not present | | | 15.5 | 3 | 3 | 60.8 | 13 | 28 |
| | | Chl4 | CENP-N | Not present | | | 7.2 | 3 | 3 | 29 | 12 | 21 |
| | | Nkp1 | | Not present | | | 26.5 | 4 | 6 | 58 | 18 | 35 |
| | | Nkp2 | | Not present | | | 15 | 2 | 3 | 35.9 | 5 | 14 |
| | Ctf3 | Mcm16 | CENP-H | 14.9 | 1 | 1 | 19.9 | 2 | 2 | 44.8 | 6 | 12 |
| | | Ctf3 | CENP-I | Not present | | | 4 | 2 | 2 | 13.9 | 12 | 19 |
| | | Mcm22 | CENP-K | Not present | | | 14.6 | 2 | 2 | 74.9 | 17 | 34 |

*Table 2 continued on next page*

*Table 2 continued*

**Table 2.** dsn1-2D assembled kinetochores

| | | | | ampC | ampC | ampC | CEN3$^{mut}$ | CEN3$^{mut}$ | CEN3$^{mut}$ | CEN3 | CEN3 | CEN3 |
|---|---|---|---|---|---|---|---|---|---|---|---|---|
| | Cnn1 | Cnn1 | CENP-T | Not present | | | Not present | | | 27.1 | 8 | 11 |
| | | Wip1 | CENP-W | Not present | | | Not present | | | 21.1 | 2 | 2 |
| | | Mhf1 | CENP-S | 48.9 | 3 | 4 | 48.9 | 4 | 9 | 21.1 | 2 | 2 |
| | | Mhf2 | CENP-X | 62.5 | 7 | 8 | 47.5 | 4 | 4 | 28.8 | 2 | 3 |
| Outer KT | Mtw1 | Mtw1 | Mis12 | Not present | | | 18.7 | 4 | 4 | 48.1 | 13 | 21 |
| | | Nnf1 | PMF1 | Not present | | | 16.9 | 2 | 2 | 30.3 | 10 | 13 |
| | | Nsl1 | Nsl1 | Not present | | | 15.3 | 2 | 2 | 69 | 15 | 21 |
| | | Dsn1 | Dsn1 | Not present | | | 13.4 | 4 | 4 | 39.2 | 21 | 31 |
| | Ndc80 | Ndc80 | HEC1 | Not present | | | 18.5 | 8 | 9 | 56.4 | 37 | 63 |
| | | Nuf2 | NUF2 | Not present | | | 15.7 | 6 | 6 | 51 | 27 | 42 |
| | | Spc24 | SPC24 | Not present | | | 38.5 | 4 | 5 | 63.4 | 12 | 26 |
| | | Spc25 | SPC25 | Not present | | | 8.1 | 1 | 1 | 37.6 | 8 | 10 |
| | Spc105 | Spc105 | KNL1 | Not present | | | 3.1 | 2 | 2 | 45.9 | 41 | 60 |
| | | Kre28 | Zwint1 | Not present | | | 3.6 | 1 | 1 | 7 | 2 | 2 |
| | Dam1 | Dam1 | | Not present | | | Not present | | | 21.6 | 5 | 6 |
| | | Dad1 | | 26.6 | 1 | 1 | 26.6 | 1 | 1 | 37.2 | 2 | 4 |
| | | Dad3 | | Not present | | | Not present | | | 29.8 | 2 | 3 |
| | | Ask1 | | Not present | | | Not present | | | 21.2 | 3 | 4 |
| | | Duo1 | | Not present | | | Not present | | | 18.2 | 4 | 4 |
| | | Hsk3 | | 15.9 | 1 | 1 | 15.9 | 1 | 1 | 15.9 | 1 | 1 |
| | | Spc19 | | Not present | | | Not present | | | 30.3 | 4 | 6 |
| | | Spc34 | | Not present | | | Not present | | | 31.5 | 6 | 11 |
| | | Dad2 | | Not present | | | Not present | | | Not present | | |
| | | Dad4 | | Not present | | | Not present | | | Not present | | |
| | MAPs | Stu2 | CHTOG | Not present | | | Not present | | | 33.6 | 23 | 31 |
| | | Bim1 | | Not present | | | Not present | | | 17.7 | 4 | 5 |
| | | Slk19 | | Not present | | | Not present | | | 1.6 | 1 | 1 |
| | | Bik1 | | Not present | | | Not present | | | 10 | 3 | 4 |

DOI: https://doi.org/10.7554/eLife.37819.012

the Mis12c pathway when the conserved Aurora B sites on Dsn1 are dephosphorylated. Although yeast cells lacking the Aurora B phosphorylation sites (*dsn1-2A*) are inviable due to low Dsn1 protein expression, mutating an additional Cdk1 site (serine 264) restores protein levels and viability (*dsn1-3A*) (*Akiyoshi et al., 2013a*; *Akiyoshi et al., 2013b*). Because *dsn1-3A* weakens the interaction between Mis12c and CENP-C$^{Mif2}$, we postulated that this linkage is not essential due to compensation by the CENP-T$^{Cnn1}$ pathway. Consistent with this, there was synthetic lethality between *dsn1-3A* and *cnn1Δ* (*Figure 7A*), indicating that CENP-T$^{Cnn1}$ becomes essential when the Mis12 pathway is misregulated. This result is similar to previous reports indicating that mutants in CENP-T/W exacerbate mutations in CENP-C$^{Mif2}$ (*Schleiffer et al., 2012*; *Hornung et al., 2014*). To further test this, we crossed *dsn1-3A* to additional mutants in the CENP-T$^{Cnn1}$ pathway (deletions of CENP-K$^{Mcm22}$ and CENP-N$^{Chl4}$). These deletions are also synthetically lethal with *dsn1-3A* (*Figure 7A*), indicating

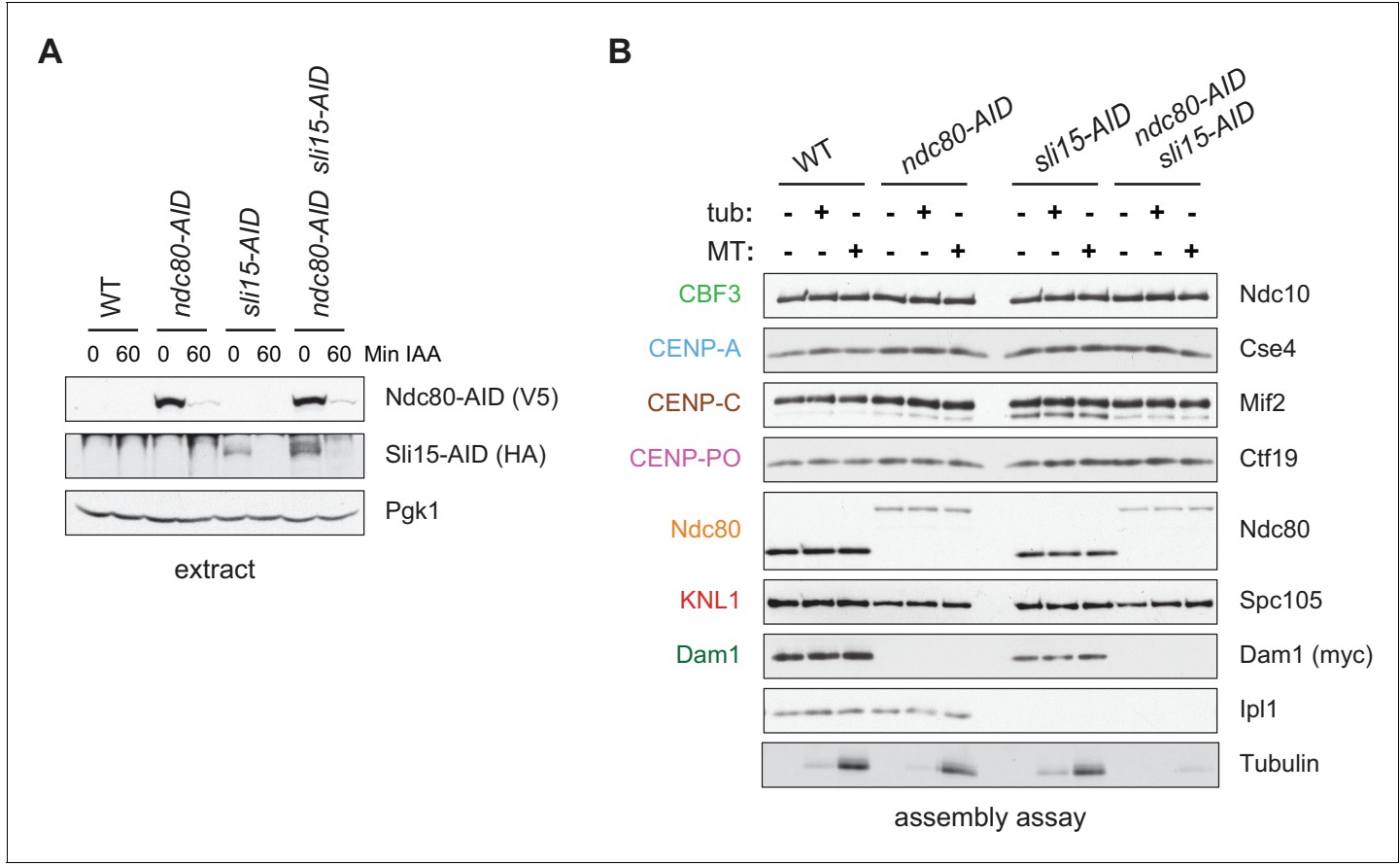

**Figure 4.** Assembled kinetochore particles bind to microtubules. (**A**) The Ndc80-3V5-AID and Sli15-3HA-AID proteins are degraded after one hour of auxin treatment as determined by immunoblotting of whole cell extracts. (**B**) Assembled kinetochores bind microtubules but not free tubulin. Assembly assays were performed using extracts from the following strains: *dsn1-2D-3Flag DAM1-9myc OsTIR1* (SBY14343), *dsn1-2D-3Flag DAM1-9myc OsTIR1 ndc80-3V5-AID* (SBY14336), *dsn1-2D-3Flag DAM1-9myc OsTIR1 sli15-3HA-AID* (SBY14890), and *dsn1-2D-3Flag DAM1-9myc OsTIR1 ndc80-3V5-AID sli15-3HA-AID* (SBY17238). All strains were arrested in benomyl and treated with auxin before harvesting. The assembled kinetochores were then incubated with buffer, free tubulin, or taxol-stabilized microtubules. The free tubulin and the microtubules contained alexa-647-labeled tubulin. DNA-bound proteins were analyzed by immunoblotting with the indicated antibodies, and the tubulin and microtubules were analyzed by fluorescence imaging. The Ndc80-3V5-AID protein migrates slower than untagged Ndc80. Extracts and tubulin input in *Figure 4—figure supplement 1*.

DOI: https://doi.org/10.7554/eLife.37819.013

The following figure supplement is available for figure 4:

**Figure supplement 1.** Whole cell extracts (left) and assembled kinetochores (right) from *Figure 4B*, immunoblotted with the indicated antibodies.

DOI: https://doi.org/10.7554/eLife.37819.014

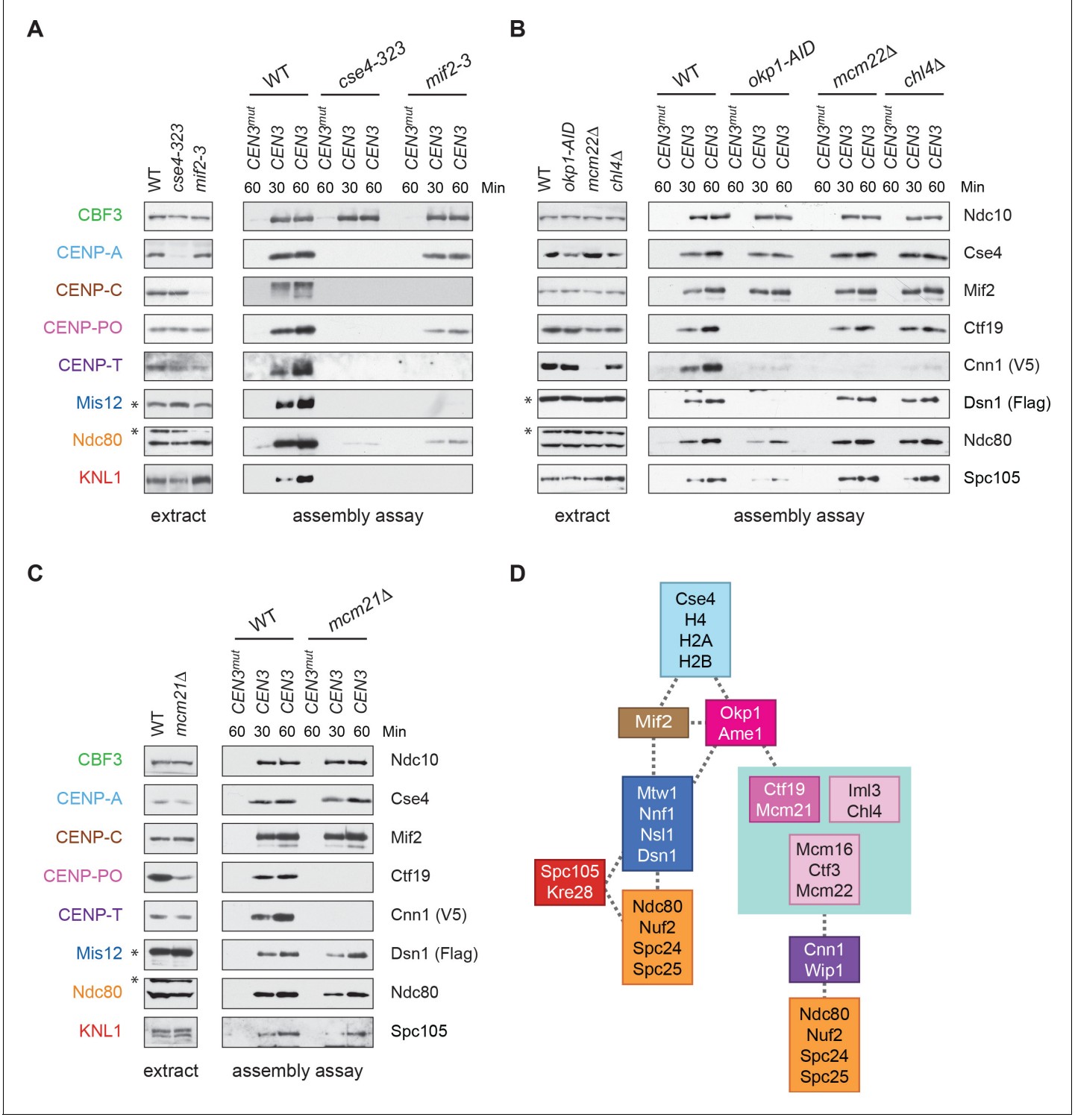

**Figure 5.** CENP-T<sup>Cnn1</sup> localization to the kinetochore requires the CCAN. (A–C) CENP-T<sup>Cnn1</sup> assembly occurs downstream of all other inner kinetochore components. Assembly assays were performed on the indicated DNA templates using extracts prepared from cells arrested in benomyl. The strains used in (A) were also shifted to the non-permissive temperature for three hours before harvesting: *DSN1-3Flag CNN1-3V5* (SBY17230), *DSN1-3Flag CNN1-3V5 cse4-323* (SBY17770), and *DSN1-3Flag CNN1-3V5 mif2-3* (SBY17603). The strains used in (B) were treated with auxin for three hours before harvesting: *DSN1-3Flag CNN1-3V5* (SBY17230), *DSN1-3Flag CNN1-3V5 okp1-3V5-AID OsTIR1* (SBY17764), *DSN1-3Flag CNN1-3V5 mcm22Δ* (SBY17460), and *DSN1-3Flag CNN1-3V5 chl4Δ* (SBY17607). The strains used in (C) were benomyl treated only: *DSN1-3Flag CNN1-3V5* (SBY17230) and *DSN1-3Flag CNN1-3V5 mcm21Δ* (SBY18304). (D) A schematic distinguishing the proteins involved in the CENP-T<sup>Cnn1</sup> and Mis12 recruitment pathways.
DOI: https://doi.org/10.7554/eLife.37819.015

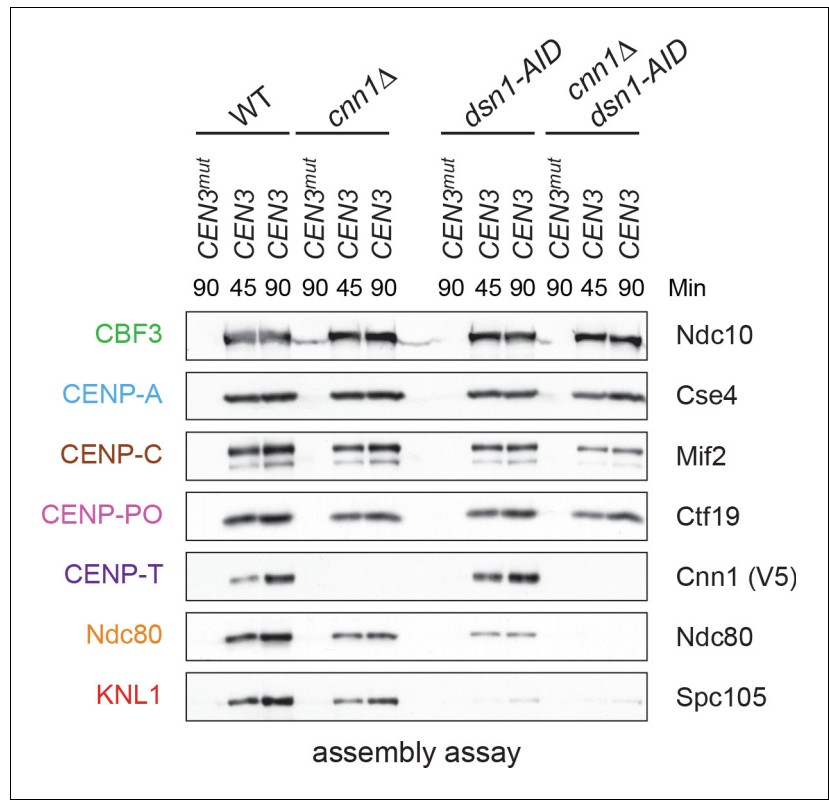

**Figure 6.** Kinetochore assembly utilizes two pathways for Ndc80 recruitment. Dsn1 and CENP-T[Cnn1] both contribute to Ndc80 recruitment. Kinetochores were assembled using extract from WT, *dsn1-AID*, *cnn1Δ*, or *dsn1-AID cnn1Δ* double mutant cells that were arrested in benomyl and treated with auxin: *DSN1-3Flag Cnn1-3V5 OsTIR1* (SBY17548), *DSN1-3Flag cnn1Δ OsTIR1* (SBY17546), *dsn1-3HA-AID Cnn1-3V5 OsTIR1* (SBY17544), and *dsn1-3HA-AID cnn1Δ OsTIR1* (SBY17380). Extracts in *Figure 6—figure supplement 1*.

DOI: https://doi.org/10.7554/eLife.37819.016

The following figure supplement is available for figure 6:

**Figure supplement 1.** Whole cell extracts (left) and assembled kinetochores (right) from *Figure 6*, immunoblotted with the indicated antibodies.

DOI: https://doi.org/10.7554/eLife.37819.017

that the entire CENP-T[Cnn1] pathway is essential for viability when the interaction between Dsn1 and CENP-C[Mif2] is crippled by a lack of Aurora B phosphorylation. These data are consistent with previous observations that a CENP-T[Cnn1] deletion has synthetic phenotypes with a CENP-C[Mif2] mutant that cannot bind to Mis12c and with a temperature sensitive allele of the Mis12 complex component *NNF1* (*Bock et al., 2012*; *Hornung et al., 2014*).

We hypothesized that the synthetic lethality exhibited in the double mutant strain is due to a defect in Ndc80 recruitment. To test this, we attempted to generate a conditional double mutant strain to analyze Ndc80 assembly. However, a *cnn1-AID* allele was hypomorphic and exhibited synthetic lethality with *dsn1-3A* even in the absence of auxin (data not shown). We therefore generated an *mcm22-AID* allele that blocks the centromere recruitment of CENP-T[Cnn1] (*Figure 7—figure supplements 1* and *2*). The *mcm22-AID dsn1-3A* double mutant was inviable when auxin was added (*Figure 7B*). To analyze Ndc80 recruitment, we performed the assembly assay from extracts made from the single and the double mutant cells treated with auxin. Because we expected the double mutant to be deficient for Ndc80 recruitment, we reasoned that the mitotic checkpoint might be compromised and therefore arrested cells in S phase rather than mitosis. As expected, the *dsn1-3A* extracts showed a considerable decrease in Ndc80 assembly as well as the other KMN components, Dsn1 and KNL1[Spc105], and that Ndc80 recruitment was further reduced in the *dsn1-3A mcm22-AID* double mutant (*Figure 7C* and *Figure 7—figure supplement 3*). Despite this reduction in Ndc80,

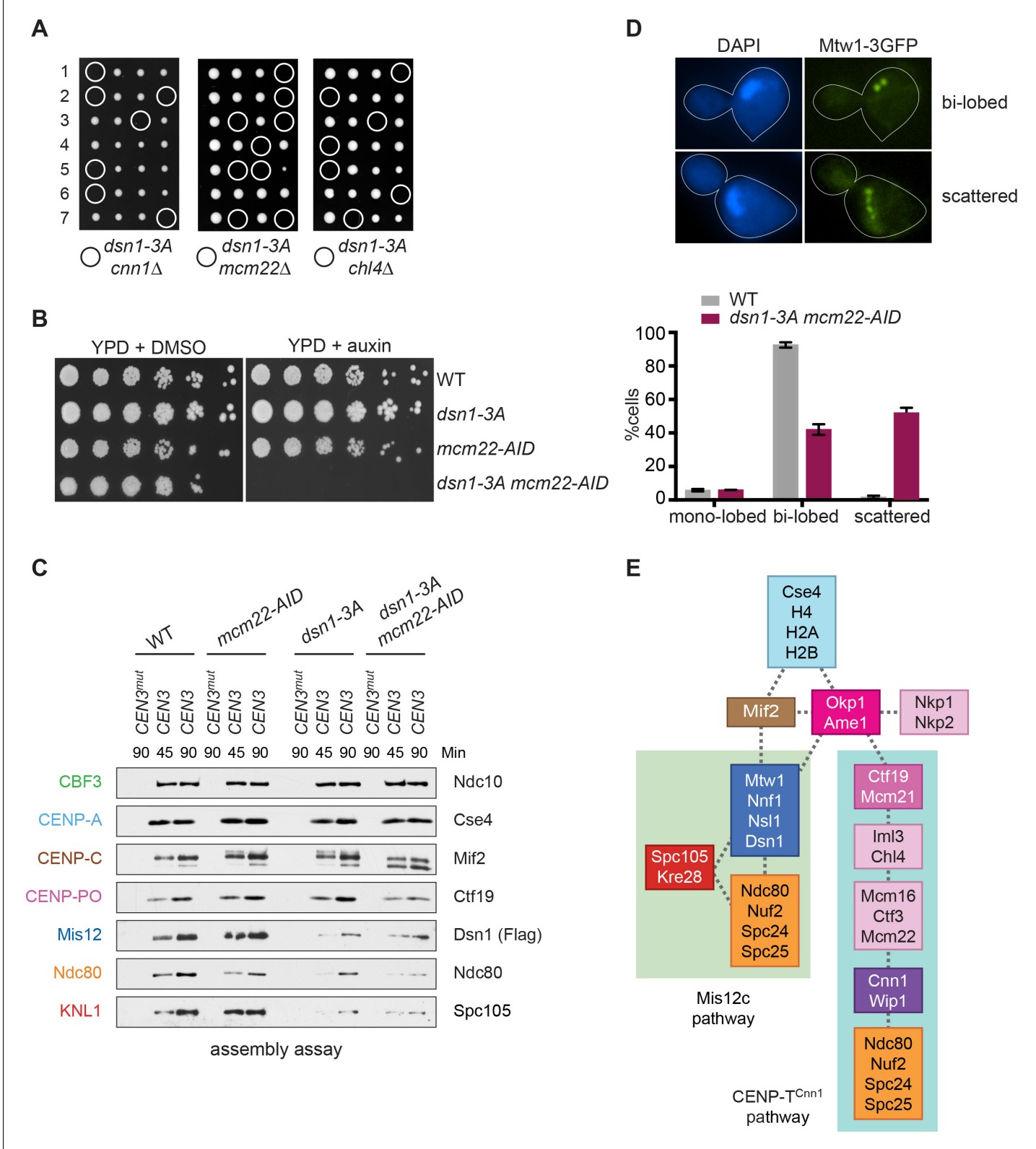

**Figure 7.** Cells require CENP-T$^{Cnn1}$ when the Mis12 pathway is impaired. (**A**) The CENP-T$^{Cnn1}$ pathway is required for viability when the Mis12c assembly pathway is compromised. *Dsn1-3A* is synthetic lethal with *cnn1Δ* and deletions of other genes (*MCM22* and *CHL4*) in the CENP-T$^{Cnn1}$ recruitment pathway. A *dsn1-3A* strain (SBY14170) was crossed to *cnn1Δ* (SBY13386), *mcm22Δ* (SBY6997), and *chl4Δ* (SBY8788). The meiotic products (tetrads) of the resulting diploids are oriented left to right, haploid spores were genotyped, and double mutants are indicated with circles. (**B**) A *dsn1-*

*Figure 7 continued on next page*

*Figure 7 continued*

*3A mcm22-AID* double mutant is lethal when treated with auxin. Serial dilutions of the following yeast strains were plated on the indicated media: WT (SBY3), *dsn1-3A-3Flag* (SBY14170), *mcm22-3HA-AID OsTIR1* (SBY17982), and *dsn1-3A-3Flag mcm22-3HA-AID OsTIR1* (SBY18171). (C) The CENP-T[Cnn1] pathway recruits Ndc80 when Mis12 complex assembly is compromised. Assembly was performed with extracts from HU-arrested strains that were treated with auxin: *DSN1-3Flag OsTIR1* (SBY14131), *DSN1-3Flag mcm22-3HA-AID OsTIR1* (SBY18044), *dsn1-3A-3Flag OsTIR1* (SBY14169), and *dsn1-3A-3Flag mcm22-3HA-AID OsTIR1* (SBY18034). Extracts in *Figure 7—figure supplement 2*. (D) WT (SBY18498) and *dsn1-3A mcm22-3HA osTIR1* (SBY18324) cells containing *MTW1-3GFP* were released from G1 and kinetochores were analyzed by fluorescence microscopy during metaphase. The percentage of cells containing mono-lobed, bi-lobed or scattered kinetochores was quantified and a representative picture of the bi-lobed and scattered categories is shown above the graph. The p value for the difference between WT and the double mutant for bi-lobed kinetochores is 0.04 and for scattered kinetochores is 0.036. (E) The sequential order of kinetochore subcomplex recruitment to the DNA, as determined from our data and from (*Pekgöz Altunkaya et al., 2016*). Dotted lines indicate physical interactions.

DOI: https://doi.org/10.7554/eLife.37819.018

The following figure supplements are available for figure 7:

**Figure supplement 1.** Mcm22-3HA-AID is efficiently degraded after 60 min of auxin treatment.

DOI: https://doi.org/10.7554/eLife.37819.019

**Figure supplement 2.** Cnn1 recruitment is blocked in Mcm22-3HA-AID strains.

DOI: https://doi.org/10.7554/eLife.37819.020

**Figure supplement 3.** Assembly assays were performed using the following strains that were arrested in hydroxyurea for two hours and then treated with IAA for one hour: WT (SBY14131), *mcm22-3HA-AID OsTIR1* (SBY18453), *dsn1-3A* (SBY14169) and *dsn1-3A mcm22-3HA-AID OsTIR1* (SBY18172).

DOI: https://doi.org/10.7554/eLife.37819.021

KNL1[Spc105] assembly appears unaffected by CENP-K[Mcm22] degradation, suggesting that the CENP-T[Cnn1] pathway may specifically recruit Ndc80 and not the full KMN network. To determine how the defect in Ndc80 recruitment we detected in vitro affects kinetochore function in vivo, we analyzed the distribution of kinetochores in the *dsn1-3A mcm22-AID* mutant. WT and *dsn1-3A mcm22-AID* cells containing Mtw1-3GFP were released from G1 and analyzed for kinetochore distribution when they were in metaphase. While the majority of WT cells exhibited a normal bi-lobed distribution of kinetochores, the kinetochores were scattered in the double mutant cells indicating a defect in establishing normal kinetochore-microtubule attachments (*Figure 7D*). Together, our data suggest that the CENP-T[Cnn1] pathway is required to recruit critical levels of Ndc80 complex to kinetochores to mediate proper kinetochore-microtubule attachments when the centromere recruitment of the Mis12c is compromised by a defect in Aurora B phosphorylation.

# Discussion

## Kinetochores can be assembled de novo

We developed an assay using centromeric DNA and whole cell yeast extracts to assemble kinetochores de novo. Although similar assays incubating yeast centromeric DNA in extracts were previously developed, none achieved assembly of the outer kinetochore (*Ohkuni and Kitagawa, 2011*; *Sandall et al., 2006*; *Sorger et al., 1994*). By altering the extract conditions, we were able to assemble all known kinetochore subcomplexes on a centromeric DNA template. Outer kinetochore assembly was dramatically enhanced when the extracts were made from cells expressing a conserved phospho-mimetic mutant that promotes kinetochore assembly in vivo (*Akiyoshi et al., 2013a*; *Kim and Yu, 2015*; *Yang et al., 2008*; *Dimitrova et al., 2016*; *Petrovic et al., 2016*). In addition, the assembly assay utilizes both conserved pathways for Ndc80 recruitment, and the assembled kinetochores are competent to attach to microtubules in vitro. In the future, it will be important to fully characterize the microtubule binding mode of the assembled kinetochores using biophysical assays.

A number of criteria indicate that the assay we developed reflects kinetochore assembly de novo. First, the assembly of all kinetochore proteins depends on the CBF3 complex, which is required to initiate assembly in vivo (*Poddar et al., 2004*). Second, CENP-A association with the template requires its chaperone (*Camahort et al., 2007*; *Shivaraju et al., 2011*; *Stoler et al., 2007*). Histone H2A, H2B, and H4 are also present, suggesting that a centromeric nucleosome forms on the DNA. We found that DNA templates capable of wrapping a single centromeric nucleosome efficiently assemble kinetochores, consistent with recent work showing that KMN can link to a single

centromeric nucleosome (*Weir et al., 2016*). In addition, these data demonstrate that pericentro-meric chromatin is not required for kinetochore assembly in vitro, although it contributes to kineto-chore function in vivo (*Bloom, 2014*). Third, outer kinetochore protein recruitment depends on the inner kinetochore proteins (*Gascoigne and Cheeseman, 2011*; *Hara and Fukagawa, 2018*). Fourth, kinetochore assembly is more efficient in extracts made from cells arrested in S phase or mitosis. Although it was previously shown that yeast kinetochores assemble during S phase, it was not clear if assembly required DNA replication (*Kitamura et al., 2007*; *Pearson et al., 2004*). Because replication cannot occur in extracts without sequential kinase treatment (*Heller et al., 2011*), our assay also shows that active DNA replication is not a strict requirement for kinetochore assembly.

## CENP-T$^{Cnn1}$ localization to kinetochores requires the CCAN

Kinetochores recruit the microtubule-binding complex Ndc80 through both the Mis12 complex and CENP-T$^{Cnn1}$ (*Nishino et al., 2013*; *Schleiffer et al., 2012*; *Malvezzi et al., 2013*; *Hori et al., 2008*). The position of CENP-T$^{Cnn1}$ within the kinetochore has been unclear because it can form a nucleo-some-like structure with CENP-W/S/X in vitro (*Nishino et al., 2012*). However, the CENP-T$^{Cnn1}$ and CENP-A$^{Cse4}$ DNA-binding sites overlap in yeast (*Pekgöz Altunkaya et al., 2016*), suggesting that CENP-T$^{Cnn1}$ may not directly contact the centromere. We found that all CCAN subcomplexes ana-lyzed are required for CENP-T$^{Cnn1}$ kinetochore localization in vitro, which is generally consistent with work analyzing its localization in human cells (*Carroll et al., 2010*; *Basilico et al., 2014*; *Samejima et al., 2015*; *Suzuki et al., 2015*; *Pekgöz Altunkaya et al., 2016*; *Logsdon et al., 2015*). However, an OA mutant that is defective in the recruitment of other CCAN components does retain some CENP-T$^{Cnn1}$ *in vivo*, suggesting that CENP-T$^{Cnn1}$ localization requirements in vivo may be more complex than revealed by our assay (*Thapa et al., 2015*; *Schmitzberger et al., 2017*). Regardless, these data are consistent with the conclusion that CENP-T$^{Cnn1}$ does not have intrinsic DNA binding activity under our assay conditions. In addition, we did not detect CENP-S/X$^{Mhf1/2}$ specifically bind-ing to centromeric DNA, suggesting they are not yeast kinetochore components.

It was previously known that CENP-A$^{Cse4}$ recruits CENP-C$^{Mif2}$ and OA, but the relative order of CCAN components downstream of these complexes was unclear. We therefore combined our data with known physical interactions of each subcomplex to map the order of the pathway from CENP-A$^{Cse4}$ to CENP-T$^{Cnn1}$ (*Figure 7D*) (*Pekgöz Altunkaya et al., 2016*). We propose that the OA com-plex is the bifurcation point of the Mis12c and CENP-T$^{Cnn1}$ assembly pathways, because the CENP-Q$^{Okp1}$ mutant perturbed the assembly of both KMN and CENP-PO$^{Ctf19-Mcm21}$, while the CCAN sub-complexes downstream of CENP-C$^{Mif2}$ and OA specifically altered only the CENP-T$^{Cnn1}$ pathway. The CENP-T$^{Cnn1}$ recruitment pathway is therefore comprised of the CENP-PO (CM complex), CENP-HIK, and CENP-LN complexes. We note that all of the non-essential, conserved yeast kinetochore proteins are specific to the CENP-T pathway, providing an explanation for why the yeast kinetochore contains both essential and non-essential proteins.

## Functions of the CENP-T$^{Cnn1}$ pathway in budding yeast

In human cells, the CENP-T pathway recruits Ndc80 both directly and indirectly. The CENP-T protein directly binds to two Ndc80 complexes and recruits a third via a phospho-regulated interaction with a Mis12 complex that is also bound to an Ndc80 complex (*Huis In 't Veld et al., 2016*; *Rago et al., 2015*). CENP-T knockdown in human cells results in severely decreased Mis12 and KNL1 complexes at kinetochores (*Gascoigne et al., 2011*; *Kim and Yu, 2015*). In contrast, we did not find evidence for the recruitment of KMN by CENP-T$^{Cnn1}$ in budding yeast, consistent with data showing that recombinant CENP-T$^{Cnn1}$ does not interact with recombinant Mis12 complex in vitro (*Schleiffer et al., 2012*). The CCAN mutants we assayed that specifically inhibit the CENP-T$^{Cnn1}$ pathway did not alter Mis12 or KNL1$^{Spc105}$ assembly. A lack of linkage between CENP-T$^{Cnn1}$ and the Mis12 complex in yeast may also explain why CENP-T$^{Cnn1}$ is non-essential and does not contribute to spindle checkpoint signaling (*Schleiffer et al., 2012*; *Bock et al., 2012*). It will be important to further analyze the relationship between the yeast Mis12 complex and CENP-T$^{Cnn1}$ in the future.

The Mis12 pathway is responsible for the majority of Ndc80 recruitment in yeast, so it is surprising that yeast cells are viable when the conserved phosphorylation sites that promote Mis12c localiza-tion are mutated (*Akiyoshi et al., 2013a*; *Kim and Yu, 2015*; *Yang et al., 2008*; *Dimitrova et al., 2016*; *Petrovic et al., 2016*). Here, we discovered these cells are viable because they use the CENP-

T$^{Cnn1}$ pathway to assemble a functional kinetochore. When the CENP-T$^{Cnn1}$ pathway is eliminated in cells lacking Dsn1 phosphorylation, Ndc80 levels are significantly reduced and kinetochores are defective in making normal attachments to microtubules in vivo. A deletion of CENP-T$^{Cnn1}$ has synthetic phenotypes with two other mutants that cripple Mis12c assembly: a CENP-C$^{Mif2}$ truncation lacking its Mis12c binding site and a temperature sensitive allele of *NNF1* in the Mis12 complex (*Bock et al., 2012*; *Hornung et al., 2014*). Taken together, these data suggest that the CENP-T$^{Cnn1}$ assembly pathway is required to recruit critical levels of Ndc80 when the function of the Mis12c pathway is reduced. CENP-T$^{Cnn1}$ kinetochore levels peak at anaphase (*Bock et al., 2012*; *Dhatchinamoorthy et al., 2017*), which is the time when Aurora B-mediated phosphorylation of kinetochore proteins is reversed by phosphatase activity. Therefore, the anaphase enrichment of CENP-T$^{Cnn1}$ might not only increase the load-bearing potential of kinetochore-microtubule attachments by recruiting more Ndc80, but also reinforce kinetochore stability when Aurora B-mediated phosphorylation of Dsn1 is removed. In addition, switching to an Ndc80-recruiting pathway that does not recruit KMN may also help silence the spindle assembly checkpoint, as KNL1 is the critical scaffold for the SAC.

The development of a kinetochore assembly assay de novo has helped to define the two pathways that assemble Ndc80 at kinetochores. Our assay is complementary to a previously developed assembly method using preassembled chromatin templates and frog egg extracts (*Guse et al., 2011*), but provides the advantage of being genetically tractable. In the future, our assembly assay will be useful for directly examining the role of other post-translational modifications in kinetochore assembly. In addition, it will provide a method to assess the biophysical and structural properties of each Ndc80 recruitment pathway to better understand how cells maintain kinetochore-microtubule attachments to ultimately ensure accurate chromosome segregation.

# Materials and methods

**Key resources table**

| Reagent type (species) or resource | Designation | Source or reference | Identifiers | Additional information |
|---|---|---|---|---|
| Gene (*S. cerevisiae*) | See *supplementary file 1* | | | |
| Strain, strain background (*Saccharomyces cerevisiae*) | W303 | | | |
| Genetic reagent (*S. cerevisiae*) | See *supplementary file 1* | | | |
| Antibody | anti-Ndc10 (rabbit polyclonal) | Desai lab | OD1 | (1:5,000) |
| Antibody | anti-Cse4 (rabbit polyclonal) | Biggins lab | 9536 | (1:500) |
| Antibody | anti-Mif2 (rabbit polyclonal) | Desai lab | OD2 | (1:6,000) |
| Antibody | anti-Ctf19 (rabbit polyclonal) | Desai lab | OD10 | (1:15,000) |
| Antibody | anti-Ndc80 (rabbit polyclonal) | Desai lab | OD4 | (1:10,000) |
| Antibody | anti-Spc105 (rabbit polyclonal) | Biggins lab | PAC4065 | (1:10,000) |
| Antibody | anti-Ipl1 (rabbit polyclonal) | Desai lab | OD121 | (1:300) |
| Antibody | anti-HA (mouse monoclonal) | Roche | 12AC5, Catalog #1-583-816 | (1:10,000) |
| Antibody | anti-V5 (mouse monoclonal) | Invitrogen | Catalog #R960-25 | (1:5,000) |
| Antibody | anti-Flag (mouse monoclonal) | Sigma-Aldrich | Catalog #F3165 | (1:3,000) |

*Continued on next page*

*Continued*

| Reagent type (species) or resource | Designation | Source or reference | Identifiers | Additional information |
|---|---|---|---|---|
| Antibody | anti-Myc (mouse monoclonal) | Covance | 9E10, Catalog #MMS-150R | (1:10,000) |
| Antibody | anti-mouse secondary (goat monoclonal) | GE Healthcare BioSciences | NA931 | (1:10,000) |
| Antibody | anti-rabbit secondary (goat monoclonal) | GE Healthcare BioSciences | NA934 | (1:10,000) |
| Recombinant DNA reagent | See *supplementary file 2* | | | |
| Sequence-based reagent | See *supplementary file 3* | | | |
| Chemical compound, drug | α-factor | United Biochemical Research Inc. | | 10 mg/mL |
| Chemical compound, drug | hydroxyurea | Sigma | H8627 | 0.2M |
| Chemical compound, drug | benomyl | Sigma | 381586–25G | 60 mg/mL |
| Chemical compound, drug | indole-3-acetic acid (IAA) | Sigma | I3750-5G-A | 500 mM |
| Other | Dynabeads M-280 Streptadivin | Invitrogen | 112-05D | |

## Yeast strain construction

The *Saccharomyces cerevisiae* strains used in this study are listed in *Supplementary file 1*. Standard genetic crosses and media were used to generate and grow yeast (*Sherman et al., 1974*). Gene deletions, *AID* alleles, and epitope tagged alleles (3Flag, 9myc, 3 HA, and 3V5) were constructed at the endogenous loci by standard PCR-based integration as described in (*Longtine et al., 1998*) and confirmed by PCR. *DSN1-3Flag*, *dsn1-2D-3Flag*, and *dsn1-3A-3Flag* were generated by PCR amplification of part of the *DSN1* gene, the Flag tags, and *URA3* using primers SB4570 and SB4571 on plasmids pSB1113, pSB1115, and pSB1142, respectively. The PCR products were transformed into yeast, and the transformants were confirmed by sequencing. The plasmids and primers used to generate strains are listed in *Supplementary file 2* and *3*.

## Yeast methods

All liquid cultures were grown in yeast peptone dextrose rich (YPD) media. Cells were arrested in G1 or S phase by adding either 10 μg/mL α-factor in DMSO or 0.2M hydroxyurea, respectively, to log phase cells in liquid culture for three hours until at least 90% of the cells were shmoos (α-factor) or large-budded (hydroxyurea). To arrest cells in mitosis, log phase cultures were diluted 1:1 with liquid media containing 60 μg/mL benomyl and grown for another three hours until at least 90% of cells were large-budded.

Temperature sensitive alleles were inactivated by diluting log phase cultures 1:1 with 37°C liquid media and shifting the cultures to 37°C for 2 hr (*ndc10-1*) or 3 hr (*cse4-323 and mif2-3*) before harvesting. For *cse4-323 and mif2-3*, the added 37°C media included 60 μg/mL benomyl.

For strains with auxin inducible degron (*AID*) alleles, all cultures used in the experiment were treated with 500 μM indole-3-acetic acid (IAA, dissolved in DMSO) for the final 60 min of growth (*scm3-AID*, *ndc80-AID*, *sli15-AID*, and *dsn1-AID*) as described in (*Nishimura et al., 2009*; *Miller et al., 2016*). For the experiment that included *okp1-AID* (*Figure 5C*), all log phase cultures were diluted 1:1 with media containing benomyl and IAA such that the final concentrations were 30 μg/mL benomyl and 500 μM IAA. After two hours, another 150 μM IAA was added, and cultures were harvested after one more hour. For the experiment in *Figure 7B*, 0.2M hydroxyurea and 500 μM IAA was added to log phase liquid cultures. After two hours, another 150 μM IAA was added, and cultures were harvested after one more hour. For the analysis of kinetochore distribution (*Figure 7D*), cells were arrested in G1 with alpha factor for three hours and IAA was added during

the final hour. The cells were washed and released into media with IAA and harvested after 100 min (when cells were in metaphase) for microscopy analysis. At least 200 cells were analyzed in duplicate biological replicates.

Growth assays were performed by diluting log phase cultures to OD600 ~ 1.0 from which a 1:5 serial dilution series was made. This series was plated on YPD plates that were top-plated with either DMSO or 500 µM IAA plates and incubated at 23°C.

## Preparation of DNA templates, Dynabeads, and competitive DNA

Plasmid pSB963 was used to generate the *ampC* and *CEN3* DNA templates and pSB972 was used to generate the *CEN3^{mut}* template used in this study. DNA templates were generated by PCR using a 5' primer with pericentromeric homology upstream of the centromere and a biotinylated 3' primer with linker DNA, an *EcoRI* restriction site, and pericentromeric homology downstream of the centromere. The latter primer was ordered from Invitrogen with a 5' biotinylation. Sequences of the primers used to PCR amplify the DNA templates are listed in *Supplementary file 3*.

The PCR product was purified using the Qiagen PCR Purification Kit and conjugated to Streptadivin-coated Dynabeads (M-280 Streptavidin, Invitrogen) for 2.5 hr at room temperature, using 1 M NaCl, 5 mM Tris-HCl (pH7.5), and 0.5 mM EDTA as the binding and washing buffer. Per 1 mg (100 µL) of beads, we conjugated 1.98 µg/mg of the 180 bp templates, 2.75 µg/mg of the 250 bp centromeric templates, or 5.5 µg/mg of the 500 bp *ampC* template to have equivalent numbers of templates on beads. After washing three times, the beads were stored in 10 mM HEPES-KOH and 1 mM EDTA at 4°C until use. Competitive DNA was made by sonicating 5 µg/mL salmon sperm DNA in dH$_2$O. The sonicated salmon sperm DNA was stored at −20°C in between uses.

## Kinetochore assembly assay

For a standard kinetochore assembly in vitro, cells were grown in 600 mL of liquid YPD media to log phase and harvested by centrifugation. All subsequent steps were performed on ice with 4°C buffers. Cells were washed once with dH$_2$O with 0.2 mM PMSF, then once with Buffer L (25 mM HEPES pH 7.6, 2 mM MgCl$_2$, 0.1 mM EDTA pH 7.6, 0.5 mM EGTA pH 7.6, 0.1 % NP-40, 175 mM K-Glutamate, and 15% Glycerol) supplemented with protease inhibitors (10 µg/ml leupeptin, 10 µg/ml pepstatin, 10 µg/ml chymostatin, 0.2 mM PMSF), and 2 mM DTT. Cells were resuspended in Buffer L according to the following calculation: (OD of culture) x (number of mL of culture harvested)=number of µL of Buffer L. This suspension was then snap frozen in liquid nitrogen by pipetting drops directly into liquid nitrogen. These dots were then lysed using a Freezer/Mill (SPEX SamplePrep), using 10 rounds that consisted of 2 min of bombarding the dots at 10 cycles per second, then cooling for 2 min. The subsequent powder was thawed on ice and clarified by centrifugation at 16,100 g for 30 min at 4°C. The resulting soluble whole cell extracts (WCE) generally have a concentration of 50–70 mg/mL. The dots, powder, and WCE were stored at −80°C if needed. 5 µL of WCE were saved in a sodium dodecyl sulfate (SDS) buffer for immunoblot analysis.

Typically, 750 µL of whole cell extract was incubated on ice for 15 min with 24.75 µg sonicated salmon sperm DNA (30-fold excess competitive DNA relative to the DNA template on beads). Then, 30 µL of beads pre-conjugated with DNA were added, and the reaction was rotated constantly at room temperature for 30–90 min. The reaction was stopped on ice by addition of 3–5 times the reaction volume of Buffer L. The beads were then washed once with 1 mL Buffer L supplemented with 33 µg/mL of competitive DNA, then three more times with 1 mL Buffer L. Bound proteins were eluted by resuspending the beads in 75 µL of SDS buffer, boiling the beads at 100°C for 3 min, and collecting the supernatant. Samples were stored at −20°C. Bound proteins were examined by immunoblotting, described below. All experiments were repeated two or more times as biological replicates to verify reproducibility and a representative result is reported.

## Mass spectrometry

Following the standard assembly protocol and washes, assembled kinetochores were washed twice with 1 mL of 50 mM HEPES pH 8, then resuspended in ~60 µL of 0.2% RapiGest SF Surfactant (Waters) in 50 mM HEPES pH 8. Proteins were eluted by gentle agitation at room temperature for 30 min. A small portion of the eluate was added to SDS buffer and analyzed by SDS-PAGE and immunoblotting and/or silver staining. The remaining sample was snap frozen in liquid nitrogen and

sent to the Taplin Mass Spectrometry Facility for LC/MS/MS analysis, or to Thermo Fisher Scientific Center for Multiplexed Proteomics at Harvard Medical School (TCMP@HMS) for TMT labeling and MS3 analysis.

## Bulk microtubule-binding assay

Microtubules were polymerized at 37°C for 15 min using a 1:50 mixture of Alexa-647-labeled and unlabeled bovine tubulin in polymerization buffer [BRB80 (80 mM PIPES, 1 mM $MgCl_2$, 1 mM EGTA, pH 6.8), 1 mM GTP, 5.7% (v/v) DMSO, and an additional 4 mM $MgCl_2$]. The polymerization was stopped with the addition of BRB80 and 10 µM taxol. Microtubules were sheared by pulling them through a 27 1/2G needle 10 times, and then pelleted by room temperature centrifugation for 10 min at 170,000 g. Polymerized microtubules were resuspended in BRB80 with 10 µM taxol to approximately 14.4 µM, based on the initial amount of tubulin. Serial dilutions of both the polymerized microtubules and the equivalent amount of initial tubulin mixture were run on an SDS-PAGE gel and analyzed by fluorescence imaging with a Typhoon Trio (GE Healthcare). The amount of tubulin that successfully polymerized was estimated to ensure that comparable amounts of free tubulin and microtubules were introduced to the assembled kinetochores. Assembled and washed kinetochores on beads were resuspended in room temperature Buffer L with 0.9 mg/mL κ-casein, 20 uM taxol, and either ~5 nM tubulin or polymerized microtubules. This reaction was incubated at room temperature with constant rotation for 45 min, then washed twice with room temperature Buffer L, resuspended in SDS buffer, and eluted by boiling. Bound tubulin or microtubules were detected by fluorescence imaging.

## Whole cell extracts for AID degradation

Whole cell extracts for immunoblotting were made by freezing cells in liquid nitrogen and resuspending in SDS buffer. Cells were lysed using glass beads and a beadbeater (Biospec Products), then clarified by centrifugation at 16,100 g for 5 min at 4°C.

## Immunological methods

Whole cell extract or samples were prepared as described above and separated by SDS-PAGE. Proteins were transferred to a nitrocellulose membrane (BioRad) and standard immunoblotting was performed. Primary and secondary antibodies were used as described in (*Miller et al., 2016*). Additionally, α-Ndc10, α-Mif2, and α-Ipl1 were generous gifts from Arshad Desai and were used as follows: α-Ndc10 (OD1) 1:5,000; α-Mif2 (OD2) 1:6,000; and α-Ipl1 (OD121) 1:300. We also used α-Cse4 (9536) 1:500 (*Pinsky et al., 2003*). HRP conjugated secondary antibodies were detected with Pierce enhanced chemiluminescent (ECL) substrate and SuperSignal West Dura and Femto ECL (ThermoFisher Scientific). Note that the immunoblot exposures vary to best represent differences across extracts or assembly samples. The levels of proteins in input extracts and assembly samples can therefore not be directly compared.

## Acknowledgements

We are grateful to Arshad Desai for generously providing many of the antibodies used in this study, and to Erman Akbay for assisting with mass spectrometry data analysis. We thank Phil Gafken and Yuko Ogata from the Fred Hutch Proteomics Facility, Ross Tomaino from the Taplin Mass Spectrometry Facility, and Ryan Kunz from TCMP@HMS for providing experimental advice and performing mass spectrometry. We thank all members of the Biggins lab for strains and insightful discussions. Finally, we thank the Biggins lab, Chip Asbury, Josh Larson and Toshi Tsukiyama for critical comments on the manuscript. The authors declare no competing financial interests.

## Additional information

### Funding

| Funder | Grant reference number | Author |
| --- | --- | --- |
| National Institutes of Health | 5R01GM064386-17 | Sue Biggins |

| Howard Hughes Medical Institute | Investigator Award | Sue Biggins |
| National Institutes of Health | Chromosome metabolism training grant | Jackie Lang |

The funders had no role in study design, data collection and interpretation, or the decision to submit the work for publication.

### Author contributions

Jackie Lang, Conceptualization, Resources, Supervision, Funding acquisition, Writing—original draft, Project administration, Writing—review and editing; Adrienne Barber, Conceptualization, Formal analysis, Validation, Investigation, Methodology; Sue Biggins, Conceptualization, Formal analysis, Funding acquisition, Validation, Investigation, Methodology, Writing—original draft, Writing—review and editing

### Author ORCIDs

Sue Biggins http://orcid.org/0000-0002-4499-6319

### Decision letter and Author response

Decision letter https://doi.org/10.7554/eLife.37819.027
Author response https://doi.org/10.7554/eLife.37819.028

## Additional files

### Supplementary files

• Supplementary file 1. Yeast strains used in this study. Complete genotypes of the *Saccharomyces cerevisiae* strains used are listed along with the strain number to reference. Replicating plasmids are indicated in brackets. All strains are isogenic with W303.
DOI: https://doi.org/10.7554/eLife.37819.022

• Supplementary file 2. Plasmids used in this study. The relevant genes and markers on each plasmid used are listed.
DOI: https://doi.org/10.7554/eLife.37819.023

• Supplementary file 3. DNA primers used in this study. The sequence and purpose of each primer used is listed.
DOI: https://doi.org/10.7554/eLife.37819.024

• Transparent reporting form
DOI: https://doi.org/10.7554/eLife.37819.025

### Data availability

All data generated during this study are included in the manuscript and supplementary data.

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
