## [Decision Letter]

Thank you for submitting your article "An assay for de novo kinetochore assembly reveals a key role for the CENP-T pathway in budding yeast" for consideration by *eLife*. Your article has been reviewed by three peer reviewers, and the evaluation has been overseen by a Reviewing Editor and Andrea Musacchio as the Senior Editor. The following individual involved in review of your submission has agreed to reveal his identity: Stefan Westermann (Reviewer #1); Hongtao Yu (Reviewer #3).

The reviewers have discussed the reviews with one another and the Reviewing Editor has drafted this decision to help you prepare a revised submission.

Summary:

In this study the authors employ a new variation of the classic "bead assay" that has been previously used in the yeast kinetochore field. They couple a short piece of centromeric DNA to beads and incubate the beads in soluble extracts prepared from yeast cells. As controls, beads coupled to either mutated CEN DNA or unrelated DNA are used. Using Western-Blotting and mass spectrometry, the authors demonstrate association of essentially all known core kinetochore proteins specifically to CEN beads. Taking advantage of existing yeast mutants they show that kinetochore assembly in this assay is abrograted in an *ndc10* mutant and requires Scm3-mediated loading of the CEN nucleosome onto the short CEN DNA stretch. They go on to use this assay to demonstrate the contribution of the Ndc80 receptor Cnn1/CENP-T to outer kinetochore assembly. In particular they demonstrate that combining a *cnn1* deletion with mutants that prevent Ipl1 mediated enhancement of Mis12 recruitment to kinetochores, abolishes Ndc80 recruitment, clarifying the contributions of these different "pathways". The major advance in this manuscript is the development of the reconstitution assay and its rigorous validation. This study, along with findings from others, establish the existence of two parallel pathways for outer kinetochore assembly: the CENP-C-Mis12c pathway regulated by Aurora B and the CENP-T-Ndc80c pathway regulated by Cdk1. The technical quality of the experiments is high. The results are convincing and compelling, even if they primarily validate findings derived from work in other systems and from recent reconstitutions with recombinant human complexes. The novelty of the work would improve if the following revisions could be implemented.

Essential revisions:

1) The authors should complement Figure 7 with quantitative analysis of Ndc80 localization using microscopy or chromatin immunoprecipitations to demonstrated the prediction that there is an additive effect on Ndc80 localization in metaphase cells in vivo.

2) The assembly assay on beads suggests Ndc80 reduction but this could be more persuasive (e.g. there is no quantification). Also, CENP-T itself is not analyzed (possibly because this requires crossing in the tagged gene). This would be a condition that would greatly benefit from the quantitative mass spec analysis used for the *dsn1-3D* mutant – specifically, a comparison of *dsn1-3A* to *dsn1-3A;mcm22-AID* by quantitative mass spec, or at least from a quantification of the signal of Ndc80 from multiple experiments.

3) The characterization of the kinetochore-microtubule binding can be strengthened. Is it possible to release the assembled kinetochore from beads and perform negative-stain EM studies on that alone or bound to microtubules? Low-resolution images would suffice. A mutant kinetochore lacking Ndc80c can be used as the negative control. We realize that this may not be straightforward, so should the authors be unable to perform this experiment, it will not be held against them. In this case, however, the authors need to add a statement clearly stating the lack of biophysical characterization of the microtubule-binding mode of the reconstituted kinetochore.

4) Subsection “Functions of the CENP-T^Cnn1^ pathway in budding yeast”, the Kim et al. (2015) study showed that depletion of CENP-T alone did not affect the spindle checkpoint in human cells. Only CENP-T-depleted cells treated with an Aurora kinase inhibitor exhibited checkpoint defect. That finding is actually in broad agreement with the current work, although the mechanism in human cells might involve Mis12c.

---

## [Author Response]

Essential revisions:1) The authors should complement Figure 7 with quantitative analysis of Ndc80 localization using microscopy or chromatin immunoprecipitations to demonstrated the prediction that there is an additive effect on Ndc80 localization in metaphase cells in vivo.

We are technically limited in our ability to quantify Ndc80 localization in vivo. While we have antibodies that work well for immunoblotting, we were not able to get them to work reliably for immunofluorescence or chromatin immunoprecipitations. While this issue could normally be resolved by epitope-tagging the Ndc80 protein, we were not able to obtain viable *mcm22-AID dsn1-3A* strains when Ndc80 was also tagged. Because we were not able to carry out the requested experiments, we further substantiated our findings in Figure 7 regarding Ndc80 levels. We have now quantified the level of Ndc80 in the assembly assay in each of the four strains reported (Figure 7—figure supplement 3) and confirmed that the double mutant strain has even lower levels of Ndc80 than the single mutant strains. Because we had difficulty directly analyzing Ndc80 levels in cells, we performed experiments to assay the phenotypes of the *mcm22-AID dsn1-3A* cells (Figure 7D). We found a significant defect in the clustering of kinetochores at the poles, consistent with defects in kinetochore function in vivo. This data is now presented in Figure 7D.

2) The assembly assay on beads suggests Ndc80 reduction but this could be more persuasive (e.g. there is no quantification). Also, CENP-T itself is not analyzed (possibly because this requires crossing in the tagged gene). This would be a condition that would greatly benefit from the quantitative mass spec analysis used for the dsn1-3D mutant – specifically, a comparison of dsn1-3A to dsn1-3A;mcm22-AID by quantitative mass spec, or at least from a quantification of the signal of Ndc80 from multiple experiments.

We were not able to analyze CENP-T levels because the *dsn1-3A mcm22-AID* cells are not viable when a tagged copy is introduced. Because we were not able to complete quantitative MS in a timely fashion, we instead quantified Ndc80 levels in the assembly assays in Figure 7 to confirm that there are lower levels in the double mutant strains as mentioned above (Figure 7—figure supplement 3).

3) The characterization of the kinetochore-microtubule binding can be strengthened. Is it possible to release the assembled kinetochore from beads and perform negative-stain EM studies on that alone or bound to microtubules? Low-resolution images would suffice. A mutant kinetochore lacking Ndc80c can be used as the negative control. We realize that this may not be straightforward, so should the authors be unable to perform this experiment, it will not be held against them. In this case, however, the authors need to add a statement clearly stating the lack of biophysical characterization of the microtubule-binding mode of the reconstituted kinetochore.

As suggested, we have now worked out conditions to cleave the assembled kinetochores from beads in preparation for functional and biophysical studies. However, we have not yet been able to obtain EM images of the assembled kinetochores because it takes a substantial amount of time and it is not our expertise so requires additional collaborators. We have visualized the isolated kinetochores bound to microtubules by TIRF microscopy. However, we have not yet carefully determined the contributions of Ndc80 due to the amount of work involved in TIRF assays, so we did not add this to the paper. Because the single molecule studies are labor intensive, our plan is that they will be part of a future publication. As requested, we have clearly stated that this is a critical goal for the future.

4) Subsection “Functions of the CENP-T^Cnn1^ pathway in budding yeast”, the Kim et al. (2015) study showed that depletion of CENP-T alone did not affect the spindle checkpoint in human cells. Only CENP-T-depleted cells treated with an Aurora kinase inhibitor exhibited checkpoint defect. That finding is actually in broad agreement with the current work, although the mechanism in human cells might involve Mis12c.

We thank the reviewers for noting this and have removed the statement that CENP-T depletion in human cells leads to a defect in the spindle checkpoint.